# Persistent and multiclonal malaria parasite dynamics despite extended artemether-lumefantrine treatment in children

Justin Goodwin [1,2], Richard Kajubi[3], Kaicheng Wang[4], Fangyong Li [4], Martina Wade[1], Francis Orukan[3], Liusheng Huang [5], Meghan Whalen[5], Francesca T. Aweeka[5], Norah Mwebaza[3,6] & Sunil Parikh [1,2] ✉

Standard diagnostics used in longitudinal antimalarial studies are unable to characterize the complexity of submicroscopic parasite dynamics, particularly in high transmission settings. We use molecular markers and amplicon sequencing to characterize post-treatment stage-specific malaria parasite dynamics during a 42 day randomized trial of 3- versus 5 day artemether-lumefantrine in 303 children with and without HIV (ClinicalTrials.gov number NCT03453840). The prevalence of parasite-derived 18S rRNA is >70% in children throughout follow-up, and the ring-stage marker SBP1 is detectable in over 15% of children on day 14 despite effective treatment. We find that the extended regimen significantly lowers the risk of recurrent ring-stage parasitemia compared to the standard 3 day regimen, and that higher day 7 lumefantrine concentrations decrease the probability of ring-stage parasites in the early post-treatment period. Longitudinal amplicon sequencing reveals remarkably dynamic patterns of multiclonal infections that include new and persistent clones in both the early post-treatment and later time periods. Our data indicate that post-treatment parasite dynamics are highly complex despite efficacious therapy, findings that will inform strategies to optimize regimens in the face of emerging partial artemisinin resistance in Africa.

Malaria is a leading cause of morbidity and mortality with an estimated 249 million cases and 608,000 deaths in 2022, overwhelmingly in sub-Saharan Africa, which contains some of the highest transmission sites in the world[1]. Artemisinin-based combination therapies (ACTs) are the primary treatment for uncomplicated malaria globally[2]. ACTs combine a potent short-acting artemisinin that rapidly reduces parasite densities by up to ten thousandfold per blood stage cycle, with a longer-acting partner drug to eliminate residual parasites during an extended period of monotherapy. ACT efficacy is threatened by widespread partial artemisinin and partner drug resistance in Southeast Asia, and

by the emergence of partial artemisinin resistance in sub-Saharan Africa[3–6]. Despite delayed parasite clearance, defined as persistent microscopic parasitemia at 72 h or a parasite clearance half-life ≥ 5 h, treatment efficacy with ACTs often remains above 90% due to the continued efficacy of partner drugs and higher levels of acquired immunity in regions such as Africa[5,7]. The partner drug is thought to not only protect against artemisinin resistance emergence and spread, but also provides a period of post-treatment prophylaxis against new infections—an important consideration in high transmission settings with frequent infective mosquito bites.

[1]Department of Epidemiology of Microbial Diseases, Yale School of Public Health, New Haven, CT, USA. [2]Yale School of Medicine, New Haven, CT, USA. [3]Infectious Disease Research Collaboration, Kampala, Uganda. [4]Yale Center for Analytical Sciences, Yale School of Public Health, New Haven, CT, USA. [5]University of California, San Francisco, San Francisco, CA, USA. [6]Department of Pharmacology and Therapeutics, Makerere University College of Health Sciences, Kampala, Uganda. ✉e-mail: sunil.parikh@yale.edu

Because of the intense exposure to infective mosquito bites in such settings, people are often concurrently infected with multiple genetic variants of the same species—referred to as a multiclonal infection—with the number of clones defined as the multiplicity of infection (MOI). Multiclonal infections may play a role in the spread of drug resistance and influence clinical manifestations of disease and treatment outcomes[8–10]. MOI is further thought to scale with transmission intensity, with MOI typically measured as 3–8 clones per person in high transmission settings[11–14]. Amplicon deep sequencing, molecular inversion probes, and molecular barcoding are increasingly being used to discriminate and quantify individual clones in multiclonal infections, identify drug resistant variants, and understand the genetic background and evolution of parasite populations[15–22]. However, in standard therapeutic efficacy studies, treatment efficacy has been based on monitoring recurrence using microscopy, which has a limit of detection around 50-200 parasites/µL. Detection of microscopic recurrence is typically followed by PCR using size polymorphism-based genotyping to discriminate between recrudescent and new infections[23–25]. If new clones are detected, they are presumed to have originated from liver stage parasites present at the time of treatment (as ACTs lack liver-stage activity), or inoculated through an infective mosquito bite after treatment[26]. The use of microscopy is standard in such studies; however, its low sensitivity misses the large submicroscopic burden of infection, which is known to contribute to transmission, asymptomatic and recurrent malaria, and potentially to drug resistance selection and/or spread[27–30].

Molecular methods of parasite detection that use DNA or RNA markers of parasitemia—in some cases 10,000 times more sensitive than microscopy—can be used to characterize this burden of infection[31–34]. The increased sensitivity of DNA or RNA markers has recently revealed a surprisingly complex picture of post-treatment parasite dynamics in which persistent submicroscopic parasitemia may be common after treatment with an ACT (Table S1)[35–39]. After treatment, the detection of parasite DNA can be due to circulating gametocytes, residual DNA in the absence of viable asexual parasites, or from artemisinin-induced dormancy, metabolically inactive parasites that have undergone developmental arrest[40,41]. But persistent detection of ring-stage markers that target parasite RNA may be more likely to represent viable circulating asexual parasites, and is suggestive of parasites which may have survived initial treatment[36–38]. Molecular studies of ring-stage markers have thus far been limited by small sample sizes, short follow-up duration, or treatment of asymptomatic infections only; none have integrated measures of drug pharmacokinetics with parasite dynamics[36–38].

Artemether-lumefantrine (AL) is the most widely used ACT globally and in sub-Saharan Africa, and the recent emergence of partial artemisinin resistance in Africa threatens its useful therapeutic life. In this study, we sought to understand how an extended AL regimen impacts submicroscopic parasitemia and infection complexity by characterizing total parasite, asexual ring-stage, and multiclonal parasite dynamics in a randomized trial of 3 day versus 5 day AL in HIV-infected and HIV-uninfected children in a high transmission setting in Uganda.

## Results

### 18S rRNA and SBP1 mRNA as markers of total and ring-stage *Plasmodium falciparum* parasitemia

Parasitemia—the presence of parasites in the blood—is routinely quantified using microscopy; however, quantifying the submicroscopic (subpatent) parasite burden requires more sensitive molecular methods. Here, we used 18S rRNA as a marker of total parasitemia due to its ubiquitous expression throughout the human blood-stage of the parasite life cycle. 18S rRNA-determined parasite densities were highly correlated with microscopic parasite densities, and could quantify parasites down to 5–50 parasites/mL ($R = 0.73$, $P < 0.001$; Fig. S1 and

Fig. S2). To further discriminate asexual ring-stage parasites from total parasites, we used skeleton-binding protein 1 (SBP1) mRNA as a marker of ring-stage parasitemia; likewise, SBP1-determined parasite densities were highly correlated with microscopic parasite densities ($R = 0.74$, $P < 0.001$), and with 18S densities ($R = 0.92$, $P < 0.001$; Fig. S2). Therefore, we refer to the 18S determined prevalence and density as total parasitemia, and the SBP1 determined prevalence and density as ring-stage parasitemia.

### Three out of four children had detectable 18S rRNA throughout 6 weeks of follow-up despite efficacious antimalarial therapy

We assessed the prevalence and density of total parasitemia in 303 children after 3 ($n = 153$) or 5 ($n = 150$) days of twice daily AL treatment. Study details and patient characteristics are summarized in Table S2. In our high transmission setting, AL retained excellent therapeutic efficacy with 100% of children microscopy negative on day 7. However, >70% of children (205/291) presented with recurrent microscopic parasitemia by 42 days of follow-up. In contrast to microscopy, we found a high prevalence of 18S parasitemia at all time points from day 7 to day 42, ranging from 65–78% of all children positive for 18S rRNA throughout the entire duration of follow-up (Fig. 1A). At presentation, the median pre-treatment 18S parasite density was $2.5 \times 10^6$ parasites/mL (2,500 parasites/µL), which decreased to 300–500 parasites/mL (0.3–0.5 parasites/µL) on days 7 and 14 (Fig. 1B), a nearly 10,000-fold reduction in the total parasite density within the first 2 weeks after treatment and well below the limits of detection for microscopy, rapid diagnostic tests (RDTs), or conventional PCR-based studies[25].

### Nearly one in six children had detectable ring-stage parasites 2 weeks after either AL treatment regimen

Next, we examined the prevalence and density of ring-stage parasites. Following treatment, we found that 8% (22/286) and 15% (43/283) of children had detectable SBP1 mRNA on days 7 and 14, respectively (Fig. 1A). The prevalence of SBP1-determined ring-stage parasitemia was consistently higher than microscopic parasitemia throughout follow-up; overall, microscopy failed to detect 39% (215/553) of the total prevalence of ring-stage parasites (94% (61/65) in the first 2 weeks after treatment). The median pre-treatment SBP1 parasite density was $5 \times 10^6$ parasites/mL (5000 parasites/µL). On days 7 and 14, the median SBP-1 parasite density was ~10,000 parasites/mL (10 parasites/µL), reflecting the superior sensitivity of SBP1 mRNA as compared to microscopy and conventional RDTs, but inferior sensitivity to 18S rRNA (Fig. 1B).

### Molecular markers reveal overestimates of microscopy-based assessments of therapeutic efficacy and post-treatment prophylaxis

Most therapeutic efficacy trials, including our own, use microscopy to monitor parasite clearance and recurrence throughout follow-up. To capture the burden of parasitemia that was potentially missed with this method of detection, we looked at the subset of participants who would be classified as having adequate clinical and parasitological response (ACPR) using WHO criteria (i.e., no evidence of parasites on microscopy throughout follow-up). Among children classified as having ACPR, up to 58% had detectable 18S rRNA, and up to 18% had detectable SBP1 mRNA from day 14 onwards throughout follow-up (Fig. 1C). There was no difference in the baseline parasite density between children who were classified as ACPR or who were found to have microscopic recurrence. Notably, microscopy was negative in 17/85 (20% of all ACPR) and 16/85 children (19% of all ACPR) who had 18S or SBP1 parasite densities >100 parasites/µL, respectively—and three who had >10,000 parasites/µL (Fig. 1D). Thus, one out of five children classified as ACPR had recurrent parasitemia that was missed by expert microscopists despite parasite densities above conventional microscopic thresholds.

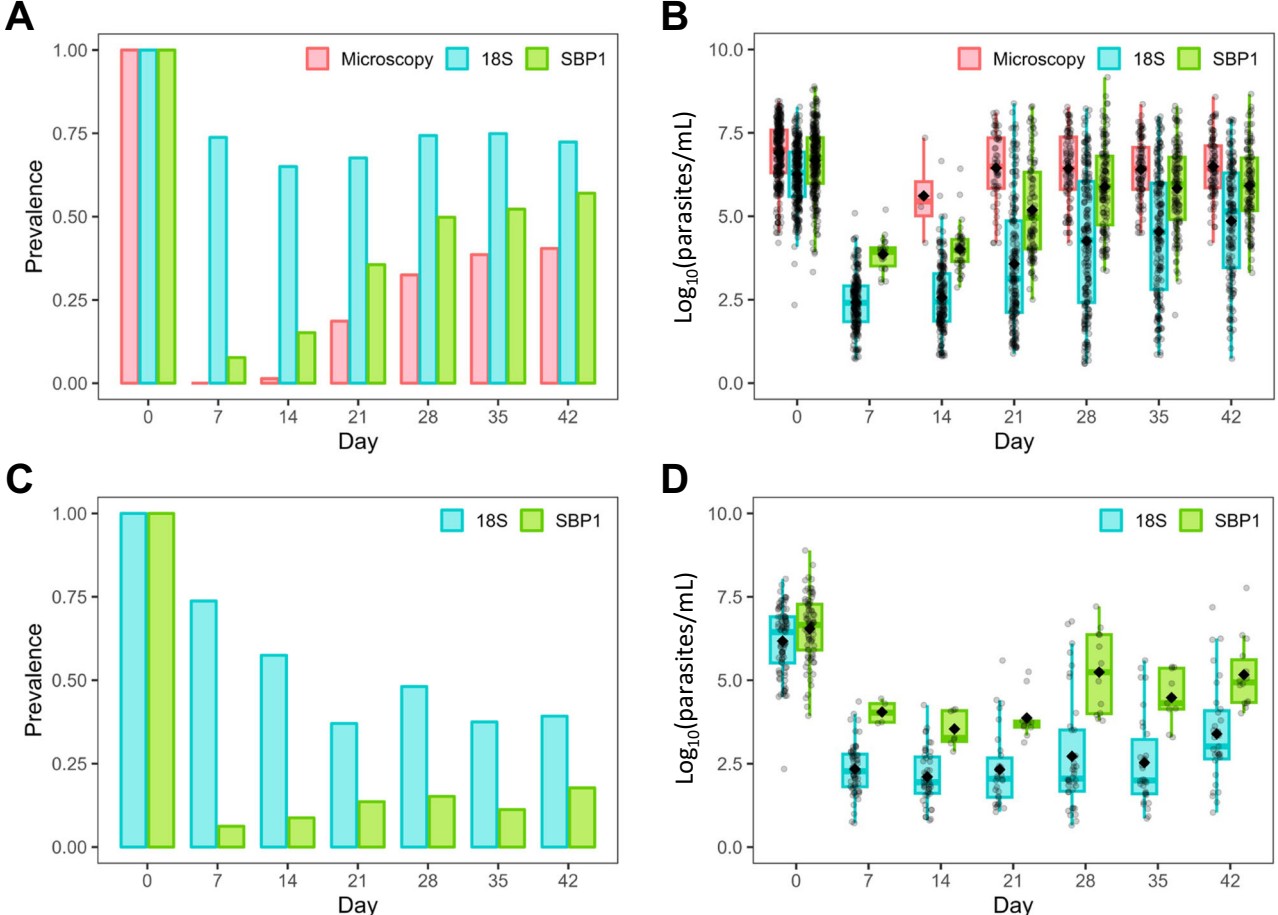

**Fig. 1 | Parasite prevalence and density after treatment with standard (3 day) or extended (5 day) AL. A** Parasite prevalence of microscopy, 18S, or SBP1 determined parasitemia after AL treatment. **B** Parasite densities (log₁₀ transformed parasites/mL) of microscopy, 18S, or SBP1 determined parasitemia after AL treatment. Sample sizes on days 0, 7 14, 21, 28, 35, and 42: microscopy *n* = 303, 290, 288, 290, 274, 254, 230; 18S n = 293, 286, 283, 284, 265, 247, 221; SBP1 *n* = 273, 286, 283, 284, 265, 247, 221. **C** Parasite prevalence of 18S and SBP1 determined parasitemia in children classified as having adequate clinical and parasitological response (ACPR). (**D**) Parasite densities of 18S and SBP1 determined parasitemia in children with

ACPR. Sample sizes on days 0, 7 14, 21, 28, 35, and 42: 18S *n* = 85, 80, 80, 81, 79, 80, 79; SBP1 *n* = 79, 80, 80, 81, 79, 80, 79. Parasite densities are presented as log₁₀ transformed parasites/mL. Box and whisker plots represent the median, first and third quartiles, and minimum and maximum values; black diamonds represent the mean. Limit of detection of microscopy = 50-100 parasites/μL. Limit of detection of 18S rRNA = 0.005–0.050 parasites/μL. Limit of detection of SBP1 mRNA = 0.1–1.0 parasites/μL. ACPR is defined as an absence of microscopic parasitemia irrespective of fever throughout follow-up. Source data are provided as a Source Data file.

## Extended AL treatment regimen reduces the rate of *recurrent* ring-stage parasitemia

We previously showed that extending AL from a 3 day to a 5 day regimen significantly improved artemether and lumefantrine drug exposure in children[42]. Despite this improved exposure, extending the duration (and number of doses) of AL was unable to significantly reduce the risk of recurrent parasitemia as defined by microscopy. We hypothesized that a 5 day AL regimen would significantly reduce molecularly-determined recurrent parasitemia (as defined by 18S or SBP1) compared to a 3 day AL regimen. We found that while 18S-determined recurrence rates were not significantly different between 3-day and 5 day AL regimens (Table 1 and Fig. 2B), SBP1-detemined recurrence rates were significantly lower during the 14–21, 21–28, and 28–35 day intervals in the extended 5 day regimen as compared to the 3 day regimen (Table 1 and Fig. 2C). Likewise, children in the 5 day arm had a 31% reduced hazard of recurrent ring-stage parasites within the first 28 days after treatment, after adjusting for age, sex, HIV status, and baseline parasite density (*P* = 0.014), as compared to those in the 3 day arm. HIV status did not significantly impact recurrence rates throughout the follow-up period by any measure of parasitemia.

Next, we examined the impact of the extended 5-day AL regimen on parasite densities throughout follow-up. Based on all three

measures of parasitemia, parasite densities appeared to be lower in the 5 day regimen compared to the 3 day regimen within the first 28 days after treatment–the expected duration of lumefantrine activity given its half-life–but only parasite densities on day 21 were significantly lower (Table S3 and Fig. 3). HIV status was not significantly associated with parasite density and thus was not adjusted for in the final model.

## The extended AL regimen and increased lumefantrine exposure are associated with decreased *persistent* ring-stage parasitemia after treatment

Previous studies have demonstrated the persistence of ring-stage parasites up to 14 days after AL treatment of symptomatic malaria[36,37]. Given that residual parasites would be exposed to the maximum effective concentrations of partner drug throughout this period, we reasoned that persistent ring-stage parasites would be associated with reduced lumefantrine exposure. We found that a higher baseline parasite density, but not treatment regimen, was significantly associated with the presence of ring-stage parasites on day 7 after adjusting for age, sex, weight, and HIV status (*P* = 0.036). In contrast, treatment regimen, but not baseline parasite density, was significantly associated with ring-stage parasitemia on day 14 after adjusting for other variables

**Table 1 | Life-table estimation of recurrence rate by treatment group**

| Intervals | Microscopy | | | 18S rRNA | | | SBP1 mRNA | | |
|---|---|---|---|---|---|---|---|---|---|
| | Failure Rate % (95% CI) | | $P^†$ | Failure Rate % (95% CI) | | $P^†$ | Failure Rate % (95% CI) | | $P^†$ |
| | 3 Day AL (*n* = 147) | 5 Day AL (*n* = 146) | | 3 Day AL (*n* = 149) | 5 Day AL (*n* = 140) | | 3 Day AL (*n* = 149) | 5 Day AL (*n* = 140) | |
| [0, 7) | 0 (0, 0) | 0 (0, 0) | | 0 (0, 0) | 0 (0, 0) | | 0 (0, 0) | 0 (0, 0) | |
| [7, 14) | 0 (0, 0) | 0 (0, 0) | | 74.1 (66.8, 80.8) | 72.1 (64.6, 79.3) | 0.712 | 8.2 (4.7, 13.9) | 7.1 (3.9, 12.9) | 0.745 |
| [14, 21) | 2.1 (0.7, 6.3) | 0.7 (0.1, 4.8) | 0.311 | 83.6 (77.2, 89.1) | 81.4 (74.6, 87.4) | 0.624 | 26.1 (19.7, 34.1) | 15.0 (10.1, 22.1) | 0.019 |
| [21, 28) | 22.1 (16.2, 29.7) | 15.8 (10.8, 22.7) | 0.168 | 89.1 (83.4, 93.5) | 90.7 (85.2, 94.8) | 0.646 | 49.1 (41.3, 57.5) | 36.5 (29.1, 45.1) | 0.031 |
| [28, 35) | 46.3 (38.6, 54.8) | 37.1 (29.8, 45.5) | 0.109 | 92.6 (87.6, 96.1) | 95.7 (91.4, 98.2) | 0.261 | 71.2 (63.6, 78.4) | 58.9 (50.9, 67.1) | 0.029 |
| [35, 42] | 63.0 (55.2, 70.9) | 54.5 (46.6, 62.7) | 0.137 | 97.0 (93.2, 99.0) | 97.1 (93.3, 99.1) | 0.960 | 78.5 (71.4, 84.8) | 74.2 (66.7, 81.1) | 0.397 |
| $HR^‡$, D28 | Ref | 0.73 (0.51, 1.05) | 0.086 | Ref | 1.01 (0.70, 1.44) | 0.976 | Ref | 0.69 (0.51, 0.93) | 0.014 |
| $HR^‡$, D42 | Ref | 0.80 (0.61, 1.06) | 0.117 | Ref | 0.90 (0.72, 1.11) | 0.315 | Ref | 0.81 (0.61, 1.07) | 0.140 |

†Comparing point failure rates using z-test.
‡Multivariable Cox regression controlling for age, sex, weight, HIV status, and baseline parasite density. Standard errors were estimated using robust sandwich estimator. Sample sizes for microscopy, 18S, and SBP1 were 289, 280 and 261, respectively. HIV status was not significant in all three models (HR 0.98 95% CI [0.74, 1.29]; HR 1.01 95% CI [0.76, 1.32]; HR 0.83 95% CI [0.54, 1.28]).

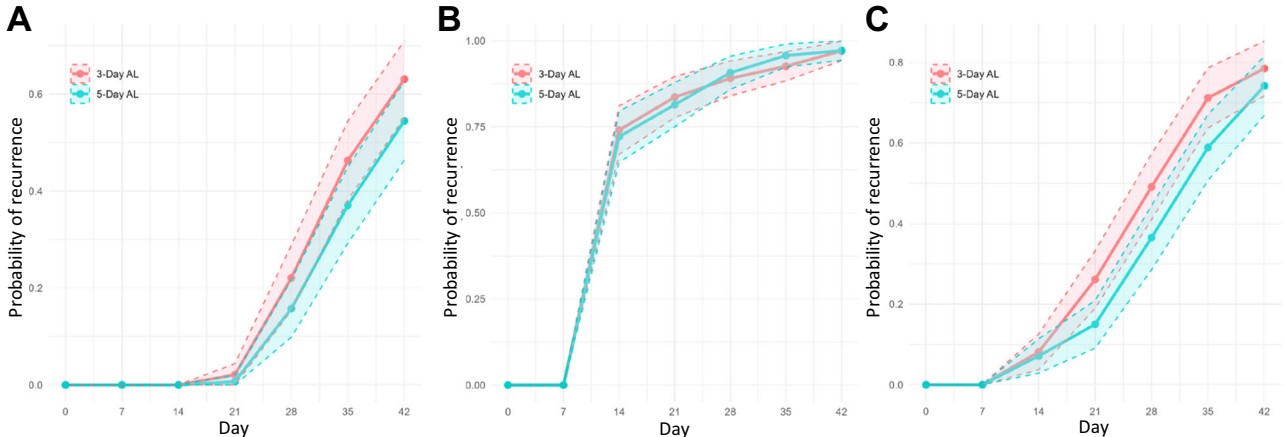

**Fig. 2 | Parasite recurrence rate after treatment with standard (3 day) or extended (5 day) AL.** Recurrence rates determined by (**A**) microscopy, (**B**) 18S, or (**C**) SBP1 parasitemia using life table estimation, with differences between treatment groups assessed using *z*-tests to compare point failure rates across each interval. The probability of recurrence is shown with 95% confidence intervals (colored shaded regions bounded by dashed lines). Sample sizes for microscopy, 18S, and SBP1 were 293, 289 and 289, respectively. Source data are provided as a Source Data file.

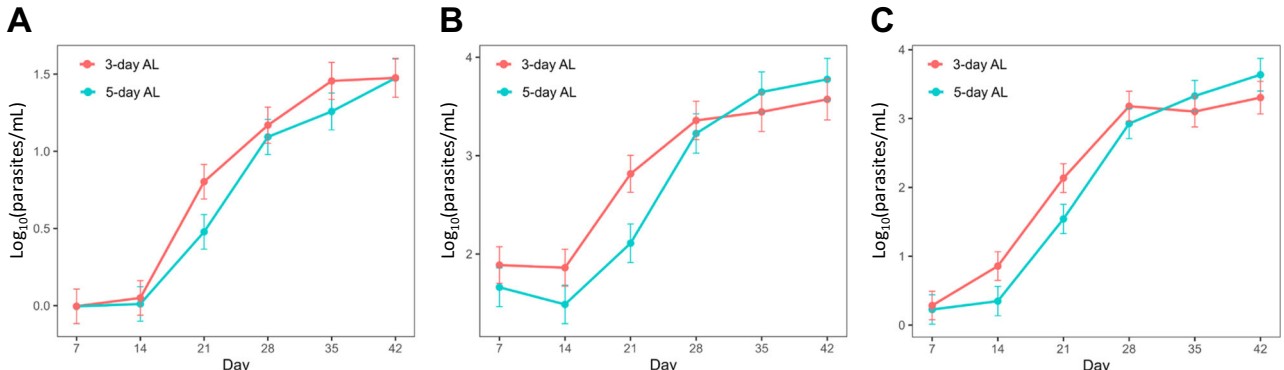

**Fig. 3 | Estimated parasite densities after treatment based on standard (3 day) or extended (5 day) AL.** Least squares means estimated parasite densities determined by (**A**) microscopy, (**B**) 18S, or (**C**) SBP1 parasitemia. Sample sizes for microscopy, 18S, and SBP1 were 303, 273 and 293, respectively. Error bars represent the standard error of the mean. Estimates are adjusted for treatment regimen, time as categorical variable, an interaction product of treatment group and time, and $\log_{10}$ transformed baseline parasite density. The *p*-values for HIV status and the interaction between HIV status and AL group were not significant; therefore, they were not adjusted for in the final model. *P*-values are derived from two-sided pairwise contrasts between treatment regimens at each time point. Source data are provided as a Source Data file.

**A**

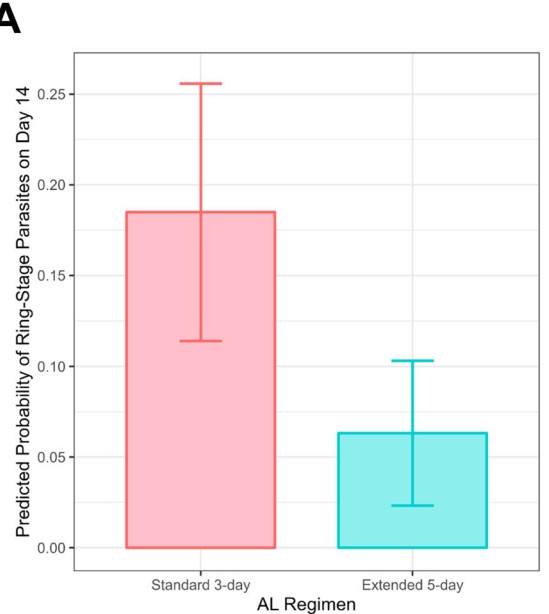

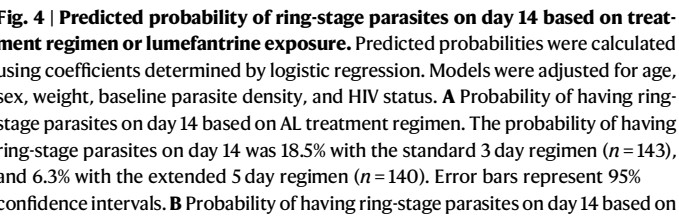

**B**

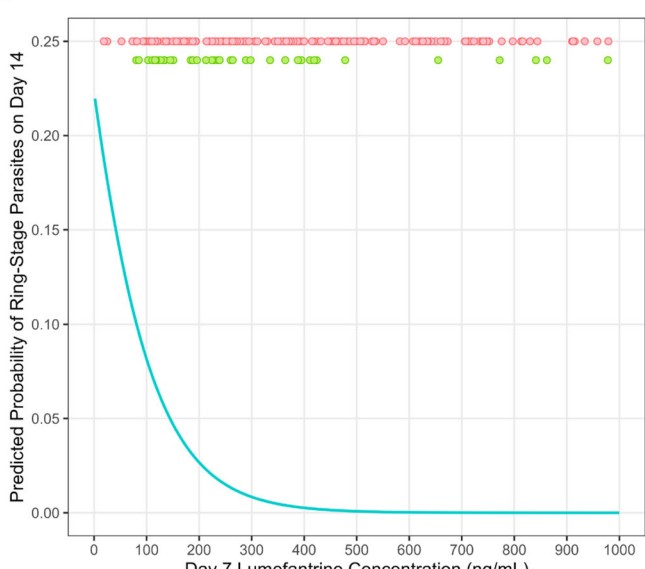

**Fig. 4 | Predicted probability of ring-stage parasites on day 14 based on treatment regimen or lumefantrine exposure.** Predicted probabilities were calculated using coefficients determined by logistic regression. Models were adjusted for age, sex, weight, baseline parasite density, and HIV status. **A** Probability of having ring-stage parasites on day 14 based on AL treatment regimen. The probability of having ring-stage parasites on day 14 was 18.5% with the standard 3 day regimen ($n = 143$), and 6.3% with the extended 5 day regimen ($n = 140$). Error bars represent 95% confidence intervals. **B** Probability of having ring-stage parasites on day 14 based on day 7 lumefantrine concentrations. Red colored points represent the day 7 lumefantrine concentrations of children who had no detectable ring-stage parasites on day 14 ($n = 179$). Green colored points represent the day 7 lumefantrine concentrations of children who had ring-stage parasites detected on day 14 ($n = 43$). 79 children without day 14 ring-stage parasites were excluded from the plot (60 with lumefantrine above 1000 ng/mL and 19 with missing data). Source data are provided as a Source Data file.

($P < 0.005$). The predicted probability of having ring-stage parasites on day 14 was 65.9% lower for children who received 5 days of AL ($n = 153$; 6.3% probability) compared to children who received the standard 3 day AL regimen ($n = 150$; 18.5% probability; Fig. 4A). Next, we looked at day 7 lumefantrine concentrations, which are highly correlated with treatment efficacy. After adjusting for age, sex, weight, and HIV status, higher day 7 lumefantrine concentrations were likewise significantly associated with a decreased probability of ring-stage parasites on day 14 ($P = 0.011$). In our cohort, for a day 7 lumefantrine concentration of 200 ng/mL, the predicted probability of having ring-stage parasites on day 14 was 2.7% (Fig. 4B). For a child with a day 7 lumefantrine concentration of 100 ng/mL (17 children in our study), the probability of ring-stage parasites on day 14 rose to 8.2%. Consistent with our model, the median lumefantrine concentrations of children with ring-stage parasites on day 14 ($n = 43$) was 260 ng/mL (IQR 281.5 ng/mL) compared to a median of 482 ng/mL (IQR 647 ng/mL) in children without ring-stage parasites ($n = 240$; $P < 0.001$).

**Deep sequencing in this high transmission setting reveals highly complex multiclonal infections throughout 6 weeks of follow-up**
To explore multiclonal infection dynamics over the course of treatment and recurrence, we used amplicon deep sequencing of three highly polymorphic markers, *csp*, *cpp*, and *cpmp*, to determine the MOI and to track the relative and absolute densities of individual parasite clones throughout follow-up. Concordance between sequenced markers was high (Fig. S3), and the median pre-treatment MOI was 4.5 clones (IQR 4.0). After treatment, nearly all children who did not clear their infection by 18S or SBP1 had a decrease in parasite density below the limit necessary for reliable sequencing (0.3–0.5 parasites/μL; Fig. 1B). Those children with densities sufficient for sequencing showed a decrease in MOI following treatment, followed by a significant increase throughout follow-up ($P < 0.005$). This trend did not differ by treatment duration (Fig. 5A and Table S4); however, we found

that MOI increased at a slower rate in HIV-infected compared to HIV-uninfected children ($P = 0.023$; Fig. 5B). The median MOI at recurrence (defined as the first day after treatment for which sequencing data was available) was not affected by regimen duration but was significantly lower in HIV-infected compared to HIV-uninfected children (3.0 (IQR 3.7) versus 4.4 clones (IQR 3.8), respectively; $P < 0.005$). Likewise, for children who progressed to clinical failure, the median MOI was significantly different between HIV-infected and HIV-uninfected children (3.3 clones (IQR 4.6) versus 7.3 clones (IQR 5.7), respectively; $P = 0.007$).

When observing individual clones over time, we defined a new clone as a sequence variant that was not identified on a previous sampling day within a patient, and a persistent clone as a sequence variant that was identified on a previous sampling day within the same patient (Fig. S4). We noted three overarching patterns that could be used to broadly characterize multiclonal longitudinal dynamics in our high transmission setting following efficacious ACT therapy. First, 55% of children with recurrent parasitemia presented with new clones repeatedly throughout follow-up ($n = 194$ children with ≥ 2 sequenced time points), suggesting a high frequency of new infective mosquito bites and/or the emergence of liver stage parasites between sampling intervals (Fig. 6A). Second, 52% of children with asymptomatic recurrent parasitemia acquired new clones superimposed on previously present clones that had not yet been cleared, such that most recurrent infections consisted of a polyclonal mixture of new and older clones (Fig. 6B). Third, the relative abundance of clones could remain stable for several weeks or shift dramatically between clones. In some cases, a clone may remain dominant for a week or two before being replaced by another clone—either a new clone or a previously detected, persistent, low-density clone (Fig. 6C). In other cases, clones would stratify into a dominant clone with several minority clones with relative densities that would remain stable through the end of follow-up (Fig. 6D).

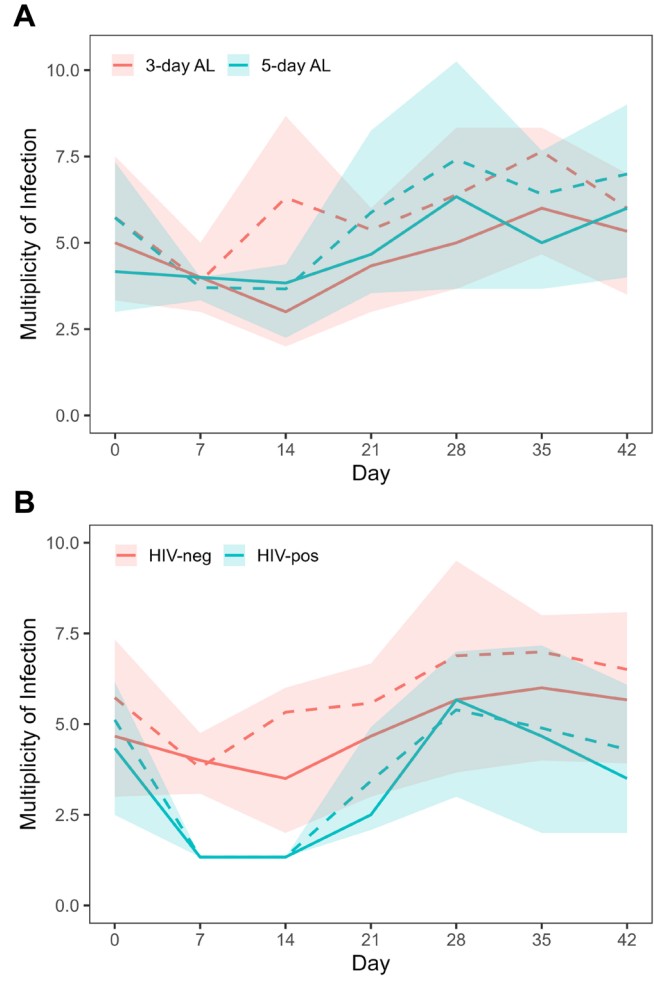

**Fig. 5 | Multiplicity of infection (MOI) over time based on (A) AL treatment regimen or (B) HIV status.** Solid lines represent the median MOI and the shaded regions represent the interquartile range. The dashed lines represent the mean MOI. Sample sizes on days 0, 7 14, 21, 28, 35, and 42 for both treatment regimen and HIV-status comparisons were 285, 19, 28, 85, 104, 122, and 108, respectively. Source data are provided as a Source Data file.

## Sequencing uncovers both residual parasitemia and early recurrences in the first 2 weeks after AL treatment

We investigated whether parasites present on days 7 and 14 represented persistent clones that were present prior to treatment, or newly infecting clones that emerged after treatment. Table S5 details the number of 18S- and SBP1-positive children on days 7 and 14. Sequencing was able to define clonal genotypes in 21 children (25 total samples with some children having positive genotypes on multiple days); 10 children had persistent clones, 10 had newly infecting clones, and one was deemed indeterminant (as sequencing failed for 2/3 pretreatment sample markers; Fig. 7). Of the 10 children with persistent clones, seven were in the 3 day AL regimen (Fig. 7A). The median day 7 lumefantrine concentration in these children was 275 ng/mL (IQR 386), and three children progressed to clinical failure (Fig. 7C). Of the 10 children with new clones detected in the first 2 weeks, three were in the 3 day AL regimen (Fig. 7B). The median day 7 lumefantrine concentration in this group was 790 ng/mL (IQR 1075), and none ended in clinical failure. Breakdown by HIV-status revealed that, in the early post-treatment period, both persistent and new clonal infections occurred overwhelmingly in the HIV-uninfected children ($P = 0.032$); only one HIV-infected child presented with persistent clones on day 7 and 14 but did not progress to clinical failure (Fig. 7D).

## Discussion

In this study, we applied molecular markers targeting different measures of parasite burden together with amplicon deep sequencing and antimalarial pharmacokinetics to investigate post-treatment parasite dynamics in a randomized trial of standard versus extended AL in children with and without HIV. Our findings reveal a striking underestimation of the longitudinal prevalence and density of parasitemia following treatment with the most widely prescribed ACT globally. Furthermore, our use of molecular markers demonstrated that while the 5 day regimen did not reduce the risk of recurrence as defined by microscopy, the extended regimen afforded a significantly lower risk of ring-stage parasitemia following treatment as compared to the 3 day regimen in our high transmission setting. This was further supported with our drug level data; day 7 lumefantrine concentrations (a surrogate predictive of clinical outcomes) were associated with a decrease in the probability of having ring-stage parasites in the early post-treatment period, providing a direct correlation between residual parasitemia and partner drug pharmacokinetics[43]. Finally, our large longitudinal and multilocus sequencing dataset revealed dynamic multiclonal patterns of new and persistent clones throughout follow-up in our high transmission setting, despite up to 5 days of clinically effective treatment.

We used 18S rRNA to estimate the total parasite burden with a higher level of precision and sensitivity compared to microscopy[44]. This global approach painted a picture of high parasite nucleic acid prevalence after treatment, with 65–78% of children having detectable 18S rRNA throughout the entire duration of 6 week follow-up. A previous study which compared the 3 versus 5 day AL regimen was conducted in a low transmission setting in Myanmar, and found a similarly high prevalence and density of 18S rDNA shortly after treatment[45]. Despite the differences in these studies (asymptomatic vs symptomatic infection, low vs high transmission), the results provide complementary insight into the impact of extended AL regimen on parasite clearance and recurrence. Notably, neither study found a difference in the prevalence or density of 18S rRNA- or rDNA-determined parasitemia between treatment regimens[45]. Whether such residual RNA or DNA represents residual nucleic acid from killed parasites is unclear; earlier in vivo studies of malaria and other parasitic infections show rapid degradation and clearance of DNA after treatment[46–48], while more recent studies show persistent detection of parasite DNA following curative treatment out to 49 days[49,50]. These discrepancies may reflect differences in the sensitivity of chosen DNA markers or differences in host immunity and/or levels of parasitemia at the time of treatment. Tun et al. conclude that a sub-population of dormant parasites are the mostly likely explanation for delayed 18S clearance in their study[45]. Artemisinin-induced dormancy is a phenomenon where a subset of asexual parasites temporarily enter a state of arrested growth and metabolic inactivity, which allows them to survive artemisinin treatment[41,51]. Such dormant parasites may be slowly killed off by partner drugs while contributing to the persistent detection of parasites after treatment.

To further elucidate the significance of the ubiquitous longitudinal detection of 18S rRNA in our study, we used the ring-stage specific marker SBP1. As anticipated, SBP1 provided additional clarity in observed post-treatment parasite dynamics. We found that 15% of children had SBP1-determined ring-stage parasitemia up to 14 days after treatment, despite up to 5 days of AL—data that are in line with two other AL studies utilizing SBP1 and another study of dihydroartemisinin-piperaquine treatment in asymptomatic children[36–38]. We further found that extending the duration of AL treatment was significantly associated with a decreased probability of ring-stage parasitemia on day 14, in contrast to 18S rRNA parasitemia. This benefit was mediated through enhanced lumefantrine exposure as assessed through day 7 lumefantrine concentrations. Notably, SBP1-estimated ring-stage parasite densities were slightly higher than the

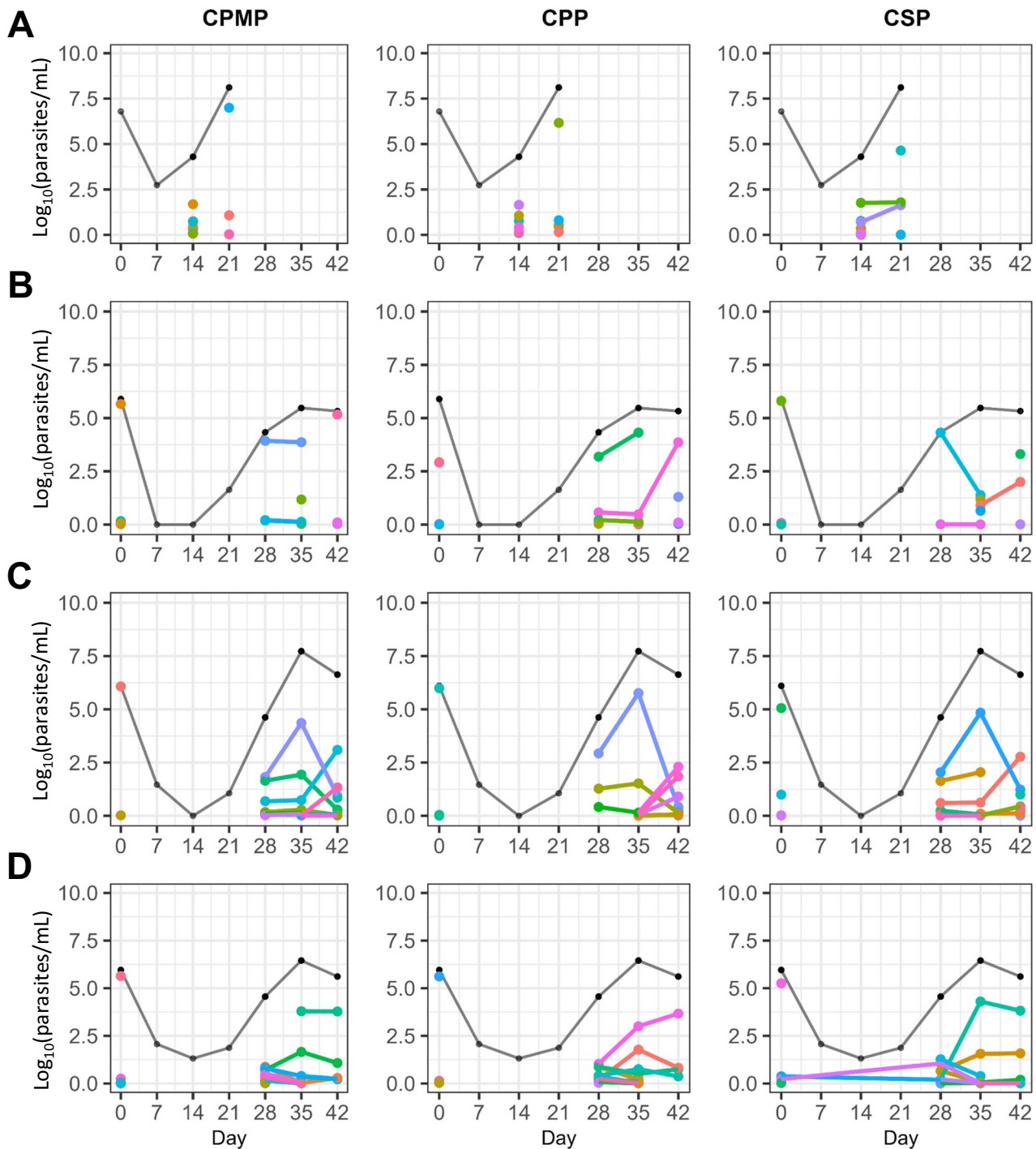

**Fig. 6 | Multiclonal infection dynamics.** The left, middle, and right columns represent *cpmp*, *cpp*, and *csp* clonal dynamics, respectively. The black line represents 18S parasite densities. **A** A child that progressed to clinical failure on day 21 presented with new clones compared to day 14. The two persistent variants identified with *csp* are discordant with *cpmp* and *cpp*, and thus likely represent different clones that share the same *csp* haplotype. Note that baseline variants are not available for this child due to sequencing failure of pretreatment samples. **B** A child with recurrent parasites presented with new clones on days 28, 35, and 42. Notably, there are persistent clones between days 28–35 (all three markers), and days 35–42 (*cpp* and *csp*), indicating a polyclonal mixture of new and older clones over time.

**C** A child with recurrent parasites presented with a multiclonal infection on day 28. By day 35 there is a single dominant clone, but on day 42 the dominant clone is overtaken by other previously detected clones (all three markers). **D** A child with recurrent parasites on day 28 displays many low densities clones with no clear dominant clone. By day 35, the infection is stratified by a dominant clone, an intermediate density clone, and multiple minority clones, with remain stable over time (at least until the end of follow-up on day 42). Note, persistent variants identified with *csp* are discordant with *cpmp*, and *cpp*, and thus likely represent different clones that share the same *csp* haplotype and not a recrudescent infection.

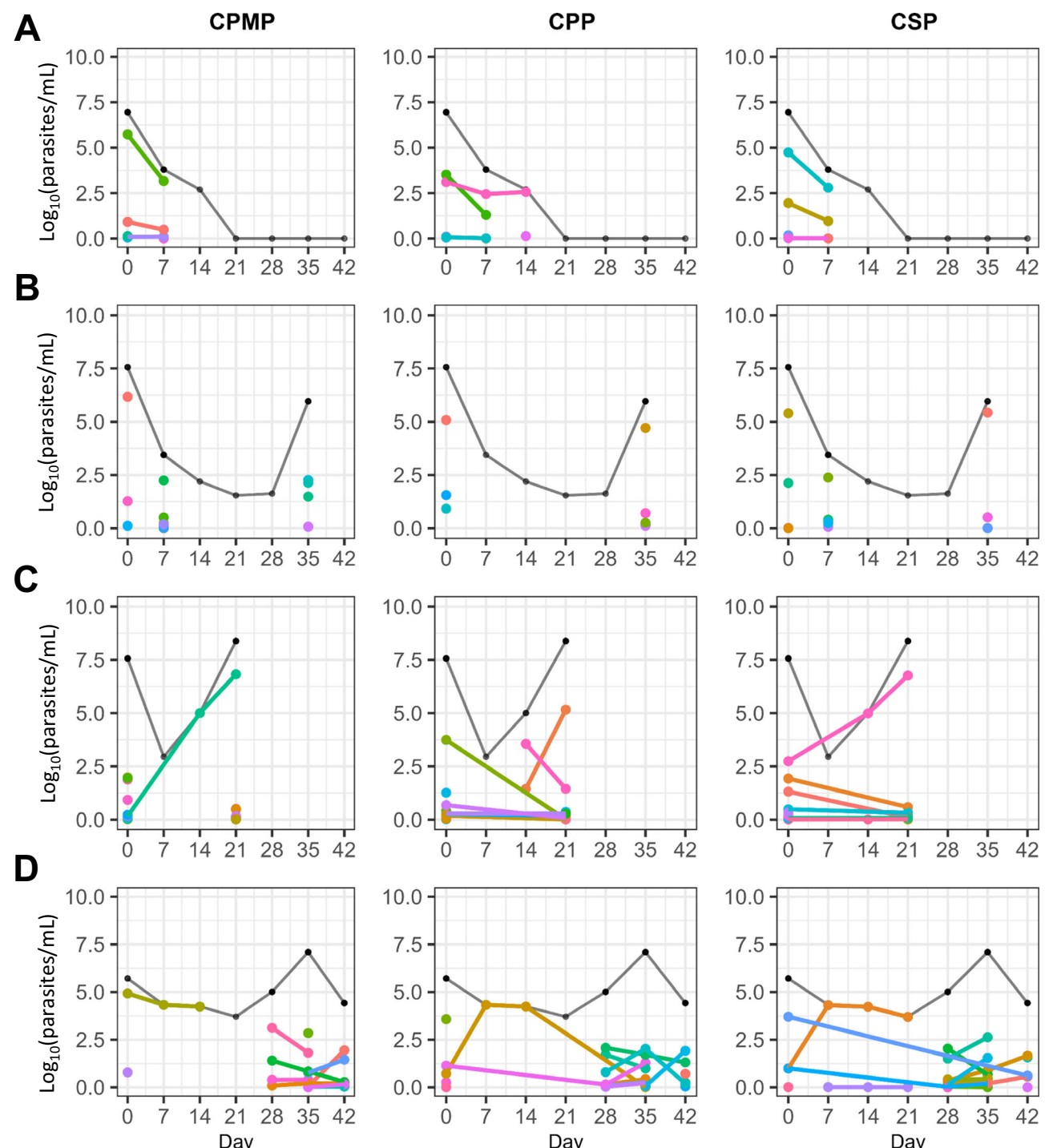

**Fig. 7 | New and persistent clonal infections in the early post-treatment period.** The left, middle, and right columns represent *cpmp*, *cpp*, and *csp* clonal dynamics, respectively. The black line represents 18S parasite densities. **A** An HIV-uninfected child with three clones that persisted to day 7 (all three markers). **B** An HIV-uninfected child with all new clones present on day 7 (*cpmp* and *csp*); none of these clones persisted and the child presented with a late clinical failure on day 35 with all new clones. **C** An HIV-uninfected child with at least one clone that persisted to day 14 (*cpmp* and *csp*) and several pre-treatment clones present at clinical failure on day 21. **D** Only one HIV-infected child presented with persistent clones, which can be found on days 7 and 14 (all three markers); many new infecting clones can be found on days 28-42 (all three markers), along with several pre-treatment clones (*cpp* and *csp*).

total 18S-estimated parasite densities, highlighting an inherent limitation of parasite quantification based on standard curves derived from a single laboratory strain—which is unlikely to capture the diversity and variable gene expression of field strains. Reassuringly, the strong correlation between our microscopy, 18S, and SBP1 estimated parasite densities demonstrates the robustness of our comparative analysis (Fig. S2). Altogether, the discrepancy between 18S and SBP1 suggests

that much of the residual 18S rRNA does not represent viable parasites. Nonetheless, ring-stage parasites demonstrate a striking and surprising persistence after ACT treatment during a time when partner drug concentrations should be effective, and extending AL regimen duration may reduce the risk of such persistence.

Next, we conducted longitudinal amplicon deep sequencing utilizing three polymorphic markers, *cpmp*, *cpp*, and *csp*, to

delineate the multiclonal dynamics over 6 weeks of follow-up in children on either regimen and with and without HIV. While the vast majority of studies only genotype pre-treatment and day of failure samples, our study demonstrated the remarkable clonal complexity of infections throughout follow-up; our pre-treatment median MOI was 4.7 clones with a surprisingly similarly high MOI (median 4.4) at the time of recurrence in our HIV-uninfected cohort. Sequencing samples weekly throughout follow-up allowed us to further track individual clones, where we found a complex pattern of recurrent parasitemia that consisted of a mix of new and persistent clones acquired over time beginning as early as 21 days after ACT treatment. Despite our ability to track and quantify individual clones over time, even our weekly sampling likely missed biologically relevant dynamics that occur on shorter time scales. Studies of asymptomatic individuals with consecutive daily sampling noted complex fluctuations in daily parasite clones and densities, reinforcing our understanding that any given sample represents only a fraction of the total parasite population[52–54]. Consistent with studies showing that size polymorphism-based genotyping typically underestimates MOI, we previously reported a pre-treatment median of 3 clones using only *msp2* in this same cohort of HIV-uninfected children[42,55–58].

In high transmission settings such as ours, differentiating between persistent and recurrent parasitemia in the first few weeks following treatment is challenging[42]. We were able to sequence 25 samples in 21 children on day 7 and day 14, which allowed us to explore clonal dynamics during this early post-treatment period. We found that approximately half of these children had new clones not detected at presentation—indicative of potential early new blood-stage infections. The remaining children had clones that had been previously detected in pre-treatment samples—indicative of potential persistent ring-stage parasites—three of whom progressed to clinical failure on days 21, 25, and 42 (versus none for new infections). Notably, most persistent clones were not detected beyond 14 days, potentially reflecting the sustained impact of lumefantrine given its half-life of 3–4 days. A notable limitation to amplicon sequencing is the limit of detection, which is comparable to other nested PCR methods (~ 1,000 parasites/mL), and the practical problem of distinguishing extremely low density clones from sequencing artifacts[14,19]. As a consequence, sequencing is unable to reliably describe the clonal dynamics of very low density infections, such as may be present in the early post-treatment period, or immediately following parasite emergence from the liver. Low-density clones present prior to treatment may have also been missed, resulting in persistent clones being misclassified as new infections. Furthermore, our data likely underestimates the true MOI, which may be biased towards higher density infections.

Our study consisted of an HIV-infected cohort of children on efavirenz-based antiretroviral therapy and daily TS prophylaxis, which is known to have antimalarial activity[2]. We previously described how HIV-infected children on efavirenz were at lower risk of recurrent malaria compared to HIV-uninfected children, despite significantly lower lumefantrine exposure, which we largely attributed to the protective effects of TS[59–61]. Surprisingly, we did not find an impact of HIV-infection (and thus TS prophylaxis) on parasite recurrence rates or densities throughout follow-up using 18S or SBP1-determined densities. This is likely attributed to the low sample size of HIV-infected children in our study, which was originally powered to detect differences in PK parameters, not parasitological parameters or treatment outcomes. Despite this limitation, amplicon sequencing revealed that HIV-infected children acquired clones at a slower rate throughout follow-up than HIV-uninfected children, and that they had fewer clones at the time of clinical failure than HIV-uninfected children, a consequence which we hypothesize is due to the protective effect of TS prophylaxis. Taken together, our data suggest that in high transmission settings, HIV-uninfected children have more complex infections

than those of HIV-infected children maintained on effective antiretroviral therapy and TS prophylaxis.

Given that ACTs have minimal gametocytocidal effects against mature gametocytes, a limitation of our study is the lack of gametocyte-specific microscopy or molecular data. It is possible that detectable 18S rRNA and amplicon sequencing markers may have arisen from mature circulating gametocytes that survived treatment[37]. In the previous 3 versus 5 day AL study in Myanmar, Tun et al. found a high prevalence of persistent 18S rDNA despite the administration of low-dose primaquine[45]. Furthermore, targeting the ring-stage specific marker SBP1 allowed us to more narrowly assess the asexual parasite burden after treatment, and previous SBP1 studies found no difference in the prevalence, density, or duration of ring-stage parasites with and without gametocytocidal drugs (primaquine or methylene blue)[36–38]. Nevertheless, additional studies assessing the specific impact of an extended AL regimen on residual gametocytemia are warranted. Another limitation of our study is the absence of drug-resistance data. Recent drug resistance surveys near our study site revealed little to no prevalence of partial artemisinin resistance-associated *kelch13* mutations[62,63]. However, it will be critical to genotype our early post-treatment samples (days 1–3) for drug resistance-associated mutations, and to compare parasite clearance dynamics using different molecular markers—especially given that extending ACT regimens is among the strategies, along with triple ACTs and multiple first-line regimens, being investigated to combat the emergence and spread of antimalarial drug resistance. Furthermore, sequencing of mutations associated with reduced partner drug susceptibility will be essential, particularly in those children who had residual day 14 SBP1-parasitemia despite day 7 lumefantrine concentrations exceeding 500 ng/mL (Fig. 4B).

New studies using molecular stage-specific and clonal parasite data, including ours, challenge previously held notions of within-host post-treatment and reinfection dynamics. We demonstrate the surprising prevalence of post-treatment parasitemia and the impact of partner drug exposure on circulating ring-stage parasites. Multilocus amplicon sequencing of persistent ring-stage parasites on day 14 further revealed that nearly half of the clones were present prior to treatment, while the rest appeared to be very early new infections. Most new clones were not detected beyond 14 days. Although this did not appear to impact clinical outcomes, parasites are only exposed to declining levels of partner drug throughout this time. Previous *msp1*, *msp2*, and microsatellite genotyping demonstrated that the overwhelming majority of microscopically recurrent parasitemia consisted of new infections, not recrudescence[42]. This suggests that the enormous burden of post-treatment parasitemia seen in our study is reflective of a high rate of new infections rather than treatment failures—and of the overall continued efficacy of AL. However, the possibility of residual ring-stage parasites, combined with early new infections, may have ramifications for the emergence and spread of ACT resistance, especially as partial artemisinin resistance currently threatens multiple African countries already facing widespread mutations conferring reduced partner drug susceptibility[4,63,64].

## Methods
### Study design and enrollment
This study uses samples collected as part of EXALT: a prospective, randomized, open-label pharmacokinetic/pharmacodynamic study of 3 day (6-dose) versus 5 day (10 dose) AL for the treatment of uncomplicated malaria in HIV-infected and HIV-uninfected children aged 0.5–18 years (Fig. S5)[42]. Enrollment and sample collection was performed between August 2018 and January 2020 at Masafu General Hospital in Busia, a high transmission region in Eastern Uganda[65,66]. AL dosing was weight-based (Coartem® Dispersible 20 mg/120 mg, Novartis, Switzerland) and administered with milk. Participants weighing < 15 kg, received 1 tablet; ≥15 – <25 kg,

2 tablets; ≥ 25 to <35 kg, 3 tablets; and ≥35 kg, 4 tablets. HIV-infected children were maintained on standard weight-based dosing of efavirenz-based antiretroviral therapy with daily TS prophylaxis. Informed consent (and assent for children <7 years of age) was provided by all children and their parents or guardians. Ethical approval was obtained at all participating institutions (Clinical-Trials.gov number NCT03453840).

## Microscopy

Thick blood slides were read and counted at the time of presentation. The parasite density of positive screening thick blood smears was estimated by counting the number of asexual parasites per 200 leukocytes, assuming a leukocyte count of 8000 /µL (or per 500 leukocytes, if the count was <10 asexual parasites/200 leukocytes). Smears were considered negative when examination of 100 high-power fields did not reveal parasites. All 2% Giemsa-stained slides were read by two highly-trained microscopists with a third read conducted in the case of discrepancies.

## Nucleic acid extraction and purification

Nucleic acids were extracted and purified from dried blood spots using a non-commercial, field-validated extraction method with a previously reported limit of detection of 20 parasites/mL[34]. Briefly, we extracted nucleic acids from 6 mm dried blood spot punches (Whatman™ 3 MM) using a guanidine thiocyanate and isopropanol-based lysis buffer for 2 h at 60 °C with vigorous shaking. Nucleic acids were purified using fritted glass fiber-based filter plates, eluted with low EDTA-TE buffer, and stored at -80 °C.

## 18S rRNA RT-PCR

To estimate the total parasite density, duplex RT-PCR was performed for *Plasmodium falciparum*-specific 18S rRNA and human actin mRNA using the QuantiTect® Multiplex RT-PCR Kit[32,34]. We used 2 µL of template in a 10 µL total reaction volume per sample, with RT-PCR performed using a Roche LightCycler 96 (LightCycler® 96 v1.1.0; Roche Diagnostics). Parasite densities were calculated using a standard curve of known parasite densities included in every RT-PCR experiment. This standard curve was created using 10-fold serial dilutions of synchronized ring-stage 3D7 parasites (range $10$-$10^{-8}$% parasitemia; $5 \times 10^8$–0.5 parasites/mL). The limit of quantification was 5-50 parasites/mL. Microscopic and 18S RT-PCR determined parasite densities were highly correlated (Fig. S1A). 18S rRNA primer and probe sequences are provided in the supplementary data.

## SBP1 mRNA RT-PCR

Skeleton binding protein 1 (SBP1) is predominantly expressed in ring-stage parasites and has been validated in laboratory and field-based studies[36–38,67]. SBP1 RT-PCR was performed using the GoTaq® 1-Step RT-qPCR System with 2 µL of template in a 10 µL total reaction volume. Ring-stage parasite densities were calculated using a standard curve of known parasite densities included in every RT-PCR experiment (see **18S rRNA RT-PCR** methods). The limit of quantification was ~1000 parasites/mL; however, the limit of detection was ~100–1000 parasites/mL, corresponding to previously reported values[36,37]. Compared to 18S RT-PCR, the sensitivity of SBP1 was 54.6% and the specificity was 97.4%. When limiting the analysis to infections with parasite densities ≥ 100 or 1000 parasites/mL, the sensitivity improved to 81.0% and 96.6%, respectively. Thus, based on the prevalence of infected patients, SBP1 RT-PCR had the ability to accurately identify ring-stage parasites in individuals with low density infections ≥ 100 or 1000 parasites/mL (positive predictive value 98.3% and negative predictive value 76.5% and 95.6%, respectively). SBP1 RT-PCR densities were highly correlated with microscopic and 18S RT-PCR densities (Fig. S1B, C). SBP1 mRNA primer sequences are provided in the supplementary data.

## Amplicon deep sequencing

Amplicon deep sequencing was used to genotype individual parasite clones using highly diverse polymorphic genes. A subset of 54 pre-treatment samples was used to assess five established markers for diversity (*ama1-D3*, *cpmp*, *csp*, *cpp*, and *msp7*)[18,58,68,69]. Over 99% (268/270) of markers were sequenced, with 90% having a read depth >10,000 (median = 21,595; IQR = 14,095). We selected *csp*, *cpp*, and *cpmp* based on discriminatory power (the total number of unique variants detected in the pre-treatment samples) and experimental robustness to genotype the remaining samples. Nested PCR was used to enrich target DNA sequences followed by PCR barcoding. Magnetic bead purification and size selection was performed before and after sample pooling. The final pooled library concentration was measured using a Qubit 4 fluorometer. Paired-end sequencing was carried out on the Illumina MiSeq platform using 2 x 300 bp chemistry with the MiSeq Reagent Kit v3. Sequencing reads were preprocessed (demultiplexing, quality filtering, trimming, and merging) using Geneious Prime version 2023 (Dotmatics) and analyzed using DADA2 version 1.18[70,71]. Samples with <100 reads or variants with a within-host haplotype frequency < 0.1% were excluded from the analysis[71]. The MOI of a given infection was calculated as the mean of the maximum number of variants identified for each marker[72]. For longitudinal tracking of clones, new and persistent clones were defined using a matching threshold of ≥ 2 markers[14]. Amplicon sequencing primer, adapter, and barcode sequences are provided in the supplementary data.

## Pharmacokinetic sampling and analysis

Pharmacokinetic (PK) sampling was divided into an intensive and sparse PK sampling schedule depending on the assigned AL treatment regimen (3 day versus 5 day), as previously described[42]. Concentrations of artemether and dihydroartemisinin were determined using liquid chromatography−tandem mass spectrometry, with a calibration range of 0.5–200 ng/mL and a lower limit of quantification of 0.5 ng/mL for both compounds, as described previously[73]. For lumefantrine, a previous quantification method was modified by reducing the plasma sample volume from 25 µL to 5 µL, and the instrument time per sample from 8 min to <2 min using a newer Waters® ultra-performance liquid chromatography column[74]. All sample and PK analysis was completed at the Drug Research Unit, University of California, San Francisco.

## Statistical analysis

Correlations between parasite densities were evaluated using Spearman's correlation. Recurrence rates were estimated using life-table analysis and the differences in failure rates between 3 day and 5 day AL regimens were compared using *z*-tests. The risk of recurrent parasitemia between treatment regimens was assessed using hazard ratios calculated by multivariate Cox regression, controlling for age, gender, weight, HIV status, and baseline parasite density. The standard errors were estimated using robust sandwich estimator to account for multiple episodes. Linear mixed effect models were used to account for repeated measures within episodes when comparing parasite density (estimated using least squares means) and MOI between treatment regimens. We applied logistic regression models to examine treatment regimen or day 7 lumefantrine concentration as possible predictors associated with the presence of ring-stage parasites on day 7 or day 14. Mann-Whitney U test was used to compare non normally distributed data between two independent groups. Models were adjusted for treatment regimen, HIV status, age, sex, baseline parasite density, and other clinically relevant parameters detailed in the primary text, as appropriate. All statistical analyses were performed using R v4.3.0 (R Core Team) or SAS v9.4 M8 (SAS Institute).

**Inclusion and ethics.** This research is part of a nearly 20 year collaboration with Ugandan colleagues that has focused on understanding

and optimizing the most widely used antimalarials in the most vulnerable populations in Uganda, including young children and those with HIV. The research is relevant locally and for the treatment of malaria throughout sub-Saharan Africa. Our collaboration is founded on principles of global health equity, with local and international researchers participating and exchanging scientific knowledge bidirectionally throughout the research process, including study design, implementation, data ownership, authorship, and presentation at scientific meetings. Roles and responsibilities of all collaborators were discussed prior to outset of the research. Ethical approval was obtained from the Uganda National Council of Science and Technology, the Makerere University School of Medicine Research Ethics Committee, the University of California, San Francisco Committee on Human Research, and the Yale University Human Investigations Committee.

### Reporting summary

Further information on research design is available in the Nature Portfolio Reporting Summary linked to this article.

## Data availability

The relevant data and code that support the findings of this study have been deposited in Figshare and can be accessed at: https://doi.org/10.6084/m9.figshare.25632636. All raw sequencing data that support the findings of this study have been deposited in the NIH NCBI Sequence Read Archive under BioProject accession number PRJNA1034496. Source data are provided with this paper.

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

## Acknowledgements

The authors are grateful to the children who participated in the original PK/PD study and their parents and guardians, and to the clinical study team and administrative staff, without whose support this study would not have been possible. We would also like to thank Myaing Nyunt (UMSOM) and Christopher Plowe (UMSOM) for generously sharing their nucleic acid extraction and 18S RT-PCR protocol, Christian Nsanzabana (Swiss TPH) for generously sharing their amplicon deep sequencing protocol, and Jillian Armstrong (YSPH) for her help with parasite culturing. The authors would especially like to thank Amy Bei (YSPH) for her guidance on amplicon sequencing, and invaluable insight throughout the development of this project. This work was supported primarily by R01 HD068174 (SP and FA), R21 HD110110 (SP), and F31 HD109060 (JG) funded by the Eunice Kennedy Shriver National Institute of Child Health and Human Development.

## Author contributions

J.G. and S.P. wrote the manuscript. J.G. and S.P. conceptualized the study. J.G. performed the experiments. M.Wade contributed reagents and logistical support. J.G., K.W., F.L., and S.P. analyzed and interpreted the data. F.A., L.H., and M.Whalen performed the pharmacokinetic analysis. R.K., F.O., L.H., M.Whalen, F.A., N.M., and S.P. conducted the clinical trial.

## Competing interests

The authors declare no competing interests.
