## [Peer Review File · Nature Communications]

Persistent and multiclonal malaria parasite dynamics despite extended artemether-lumefantrine treatment in childrenReviewers' Comments:

Reviewer #1:

Remarks to the Author:

This is a very interesting manuscript looking at the dynamics of early and late parasite clearance after treatment with highly efficacious ACT treatments, using highly sensitive diagnostic techniques. It is already known that using highly sensitive techniques provides a different outcome on parasite clearance and treatment outcome, and this has potential implications for the emergence of resistance. This study provides a comprehensive overview on the factors that could affect parasite clearance, including sub-microscopic parasitemia, drug concentration and parasite genotypes.

However, I would have liked the authors to explore more the potential implication for the emergence of resistance. Why did the authors only explore the genetic diversity and MOI, but not the resistance markers for K13 that could have an impact on early parasite clearance? And partner drug that could impact the late recurrence, especially new infections as they have already explored in their previous work: PMID 36260349 ?

I would also have liked the authors to explore more the Amplicon sequencing data. Is there a change in relative density over time for the persisting or recurrent clones ? Do you see any relationship between amplicon sequencing data and 18s or sbp1 data ?

It would be good for the authors to briefly discuss about the limitations of qPCR parasites quantification. Indeed, 18s is a multicopy gene with 5 to 8 copies. On the other side schizonts can contain 16 to 24 genomes, so the accuracy of the quantification may depend on the relative proportions of the different parasite stages in the sample. Using sbp1 transcripts has also limitations, that may improve specificity by detecting "viable" parasites, but the high number of transcripts may overestimate the real number of rings. It may be the case in this study as the median pre-treatment parasitemia by 18s (2500 parasite/ul) is lower compared to the parasitemia by sbp1 (5000 parasites/ul). This should be the other way around as 18s is supposed to detect the total parasite biomass, and sbp1, only ring stages. Using only synchronized ring stages for the standard curve may improve the accuracy of the quantification (similar 18s copies and approximately same number of sbp1 transcripts)

Based on these limitations and figure 2, could the authors speculate on which diagnostic tool would be more accurate in predicting recurrence between 18s and sbp1 assays ? Indeed the probability is between 60 and 80% for 18s, and between 5 and 15% for sbp1 at day 14.

The authors explain the lack of potential protection by TS in the HIV cohort by the small sample size, what about high level of mutations in Pfdhfr and Pfdhps in Uganda potentially associated with resistance to TS ?

I would have further minor comments

- Overall, I find that there are too many references in the results section, it is difficult sometimes to know if the authors are referring to their results or to other publications. I would rather keep the references for the discussion.

- The authors use interchangeably, the terms "persistent infection" or "recurrences". In my understanding, "persistent infections" are the ones that do not disappear over time, but recurrences may be "recrudescence infections" or "new infections". The authors should clarify this in the manuscript.

- Line 59: the reference is for South East Asia, not for sub-Saharan Africa

- Line 75: even though NGS techniques are increasingly used in malaria research, old techniques such

as agarose gels and capillary electrophoresis are still used, especially in malaria endemic settings

- Line 79: I would replace "old and new clones" by "recrudescence and new infections"

- Line 247: What does "sequence" mean? Did you use single SNP or microhaplotypes?

- Line 415: I would replace "exemplify" by "demonstrate"

Reviewer #2:

Remarks to the Author:

Justin Goodwin and colleagues have followed HIV-positive and negative individuals after treatment for *P. falciparum* malaria (with a 3 or 5 day treatment regimen). They screened samples by PCR for parasites during follow-up, used a stage-specific marker to detect ring-stage parasites, measure drug levels, and sequenced parasites during follow-up. They produced a rich dataset that can help to understand treatment failure.

Main comments

1) Given the different methods used, and several subgroups analyzed, in parts the text is difficult to follow. Sample sizes are almost never mentioned, not even in the abstract. In the results section, often only percentages are given. It would be helpful to add n/N in brackets. Given the many different groups and comparisons, sample sizes end up often relatively small, which makes interpretation of the strength of the results challenging. This is a major weakness of the writing. Following this point, a main question that arose while reading was what this study aims to show in the first place. Is it the comparison of 3d vs 5d treatment? The comparison of HIV+ vs HIV- individuals? A comparison of microscopy, 18S, and SBP1 to determine treatment failure? Clone dynamics? The manuscript would benefit greatly from a clearer focus.

Some of the truly interesting data are deeply hidden in the results and hard to extract. E.g., how many patients with parasites detected by highly sensitive PCR at days 7 and 14 progress to treatment failure, and this should future studies apply these methods to day 7 and 14 samples? And how many samples were available to address these questions?

2) 18S RT-PCR is well known to be very prone to contamination because of aerosol formation. A large number of negative controls and possibly a cutoff based on CT value is required. Could it be that some of the samples positive by 18S only were false-positive. Please provide details of controls used.

3) There is a number of claims that do not seem fully supported by the data. The interpretation of the data is further complicated by the above-mentioned lack of detail, often no P-values or other supporting test results are provided.

E.g. lines 190-192: "SBP1-determined recurrence rates were significantly lower during the 14-21, 21-28, and 28-35 day intervals in the extended 5-day regimen as compared to the 3-day regimen."

Please add numbers and a test result. Given the figure the difference seems minimal.

Lines 121-122: "Our high-resolution parasitological, clinical, and pharmacological data enabled us to characterize multiclonal infection dynamics at a depth and duration not previously reported". There is virtually no detail on pharmacology apart from a few mean values in different groups. There are certainly studies with longer follow up and sample size. This statement does not hold.

Lines 199-201: "parasite densities consistently appeared to be lower in the 5-day regimen compared to the 3-day regimen within the first 28 days after treatment." In Table 2, hardly any P-values are significant. Given how stochastic density estimates are, I am not convinced to see a clear pattern.

Figure 4B: Would it be possible to overlay the raw data (concentrations in each patient)? The curve alone provides little information.

There is a relatively strong focus in title, abstract, and manuscript on clone dynamics. I find this component of the study among the weakest. There are countless mentions of 'complex patterns'. The main finding of the genotyping is that in some patients, new clones were detected, while in others clones persisted (i.e. true treatment failure), and that many individuals carried multi-clone infections. These data reflects results of many other studies that applied genotyping. I didn't see a framework to put the genotyping results into a broader picture, despite multiple published mathematical models that do so.

Lines 247-262 describe some interesting patterns, but little detail is provided. At the least, please add some numbers on the frequency of each pattern observed. More in-depth analysis would be welcome.

The discussion states "We conducted the largest study to date of longitudinal amplicon deep sequencing". This is another bold claim. I don't have time to look up every single study and compare sample sizes, but the authors might want to look into <https://pubmed.ncbi.nlm.nih.gov/34146476/>, <https://pubmed.ncbi.nlm.nih.gov/33904907/>, <https://pubmed.ncbi.nlm.nih.gov/31819062/> and others. I don't think the statement holds.

While the authors discuss the limit of detection that allowed them to type a sample (e.g. line 287), the LOD of minority clones in multiple clone infections seems not addressed – potentially more relevant here. The possibility that clones might have been missed at baseline is not mentioned at all, despite multiple modeling approaches to address this problem. These studies should be cited and discussed (and possibly modeling approaches applied to the current data).

Line 436: "Our molecular stage-specific and clonal parasite data, the largest such dataset published to date". Of course, this is the largest study using exactly the methods used here, because every study is different. But rather than putting out such statements again and again use the space to cite and discuss previous relevant research.

4) At nearly 2500 words, the discussion is extremely long and could easily be shortened. Nearly the full introduction is repeated in the discussion.

5) The abstract states that "findings will inform strategies to optimize regimens in the face of emerging artemisinin resistance in Africa". It would be nice to discuss some of these new strategies in the discussion.

Minor comments

Line 73-74: In fact (and as mentioned on line 78), size-polymorphic markers have been the primary method to determine MOI in the past

Lines 76-84 are not correct. Many TES have used PCR for well over a decade now.

Line 94-97: Please clarify to what studies this statement refers. If the statement refers to treatment efficacy studies, many had large sample sizes and appropriate follow up periods. I do not understand the statement on treatment of asymptomatic individuals.

Line 185: Please add a reference.

Lines 192-195 seem to directly contradict lines 185-187. Or do lines 185-187 refer to a different study?

Line 366-268: I would argue follow-up is too short to make this claim.

Lines 378-379: "In high transmission settings such as ours, differentiating between persistent and recurrent parasitemia in the first few weeks following treatment is challenging." Why is this, and why does it differ by transmission intensity?

Lines 381-385: I believe the wording is wrong. 'Recurrence' can be treatment failure (=same clone) or new infection (=new clone). In line 382, did the authors mean 'early recrudescence'?

Are the references in Table S1 supposed to refer to the references in the main manuscript? They don't match.

Figure 6: Why are no clones shown for Child A at baseline? Given the high parasite density, sequencing seems feasible. Or should line 272 refer to figure 6, not 7?

Reviewer #3:

Remarks to the Author:

This is a very detailed clinical and parasitological comparison of therapeutic responses to the standard 3 day artemether-lumefantrine antimalarial drug treatment regimen and an extended 5 day course conducted in Ugandan children. The main study was published earlier this year. This is a substantial body of work. These comments (mixing trivial with potentially important) are in manuscript order.

The first sentence of the abstract is rather unwieldy

Line 51: Malaria is a leading cause of morbidity and mortality with over 247 million cases and 619,000 deaths in 2021 – could you add the word "estimated" as we know these figures are inaccurate.

Line 58: partial? Odd WHO favoured terminology, but isn't the threat just resistance?

Line 72: with MOI typically measured as 3-8 clones per person (as there are likely other lower density clones).

Line 82: "The use of microscopy is standard in such studies; however, its low sensitivity misses the large submicroscopic burden of infection, which is known to contribute to transmission, drug resistance selection, and recurrent episodes of clinical malaria". Really? What is the evidence for drug resistance selection? Clinical malaria is associated with microscopy detectable parasitaemias which are preceded by lower densities.

Line 92: "persistent detection of ring-stage markers that target parasite RNA may be more likely to represent viable circulating asexual parasites". But this could also represent the well described persistence of development arrested ring stages – which do not necessarily develop further (see Tun et al 2018: 3 days versus 5 days AL -42 days follow up-referenced below), although the measurement of skeleton-binding protein 1 (SBP1) mRNA does suggest potential viability.

Line 150: The limit of parasite detection in this study is not 20 parasites/mL. The actual limit of detection and accuracy of density estimation in the blood spots needs clarification. The calibration curve shows linearity in diluted test samples to a density below 10 parasites/mL but the samples are dried blood spots, so the limits of detection are determined by the volume of blood sampled, not the assay methodology per-se. Has there been a quantitation validation based on blood spots? More details on sample volume (and filter paper area and thickness and relationship with blood volumes) need to be provided.

Line 158: "found that 8% and 15% of children had detectable ring-stage parasites on days 7 and 14"
– No: had detectable levels of SBP1 mRNA.

Line 163: On days 7 and 14, the median SBP-1 parasite density was approximately 10,000 parasites/mL (10 parasites/ μ L) -this is very similar to Figure 2 in Tun et al (2018).

Line 180: were the blood slides rechecked? If still negative - what is your conclusion? The better way to express the relationship between the different parasite density estimation methods is by Bland-Altman plots -not by correlation derived from linear regression analysis.

Line 190: The comparison of the two molecular methods and the difference between the drug regimens is an important observation, suggesting that much of the persistent DNA is not in viable parasites? The exposure response data are also very valuable.

Line 228: Are the MOI results confounded by parasite density (i.e. the lower the density the lower the characterizable MOI)?

Line 285: "Our findings reveal a striking underestimation of the longitudinal prevalence and density of parasitemia following treatment with the most widely prescribed ACT globally."-Might be nice to acknowledge Tun et al (2018) who made the very same observation with the same randomized comparison, albeit without detection of persistent parasite mRNA. Importantly Tun et al studied patients in a low transmission setting where reinfection was not a major confounder and there were very few clinical failures. The discussion could be condensed as the CHIM data are not useful in comparison to the clinical trial data.

Line 340: "Given that ACTs have minimal gametocytocidal effects"; I think minimal is unfair – the activity is substantial. However, this leads to an important point-how do we know that the persistent DNA did not come from gametocytes. Why not look for gametocyte mRNA to answer that definitively?

Line 382: If 10 hepatic schizonts liberated their progeny within one day, they would release about 350,000 merozoites into a blood volume of, say, one litre in a young child. That is around the limit of detection in your assay. Please consider that as a possible cause of the "early reinfections". It does NOT mean drug failure.

The 7 pages of discussion are a nice read, but could easily be condensed into 3 without losing the main points.

Table 2: suggest either omit or explain it better.

Could we have some basic clinical details? fever? anaemia etc?

AL dosing was weight-based (Coartem® Dispersible 20 mg/120 mg, Novartis, Switzerland) and administered with milk. Does this mean the dosing was weight adjusted in bands or by exact weight? Were the artemether and DHA measurements used in this report?

References: Probably does not need all 88 references but you may wish to include and discuss the trial of Tun et al conducted between 2013 and 2015 in Myanmar which also compared 3 days versus 5 days AL, and also documented protracted submicroscopic parasitaemia, but without recrudescence. Tun KM, et al. Effectiveness and safety of 3 and 5 day courses of artemether-lumefantrine for the treatment of uncomplicated falciparum malaria in an area of emerging artemisinin resistance in Myanmar. *Malar J.* 2018 Jul 11;17(1):258. doi: 10.1186/s12936-018-2404-4. PMID: 29996844; PMCID: PMC6042398.

NJ White

Reviewer #1 (Remarks to the Author): This is a very interesting manuscript looking at the dynamics of early and late parasite clearance after treatment with highly efficacious ACT treatments, using highly sensitive diagnostic techniques... This study provides a comprehensive overview on the factors that could affect parasite clearance, including sub-microscopic parasitemia, drug concentration and parasite genotypes.

1) *I would have liked the authors to explore more the potential implication for the emergence of resistance. Why did the authors only explore the genetic diversity and MOI, but not the resistance markers for K13 that could have an impact on early parasite clearance? And partner drug that could impact the late recurrence, especially new infections as they have already explored in their previous work: PMID 36260349?*

The reviewer highlights a very important aspect of this work, but it is beyond the scope of this manuscript. We have recently begun to sequence candidate drug resistance markers to study this very question, as well as interrogate early post-treatment samples (taken on days 1-5) that were not presented in this manuscript. The initial amplicon-based sequencing will focus on K13 mutations as we expect these to be most relevant for clearance rates in the first 72 hours. We additionally plan to sequence *pfmdr1* and *pfprt* involved in partner drug susceptibility to construct drug resistance haplotypes, rather than look at single point mutations or loci. Finally, in line with our previous work which the reviewer referenced, we will integrate this data with artemisinin and lumefantrine PK data to develop population PK/PD models. Given the tremendous volume of work required to conduct drug resistance sequencing on all of our samples, additional molecular work on samples from days 1-5 post-treatment, as well as the creation of the PK/PD and mathematical models, we note that this is well beyond the scope of the current manuscript. However, we believe it is important to discuss this as a limitation of our study so we have revised the discussion to incorporate this (lines 381-390):

“Another limitation of our study is the absence of drug-resistance data. Recent drug resistance surveys near our study site revealed little to no prevalence of partial artemisinin resistance-associated *kelch13* mutations.^{62,63} However, it will be critical to drug resistance genotype our early post-treatment samples (days 1-3) and to compare parasite clearance dynamics using different molecular markers—especially given that extending ACT regimens is among the strategies, along with triple ACT and multiple first-line regimens, being investigated to combat the emergence and spread of antimalarial drug resistance. Furthermore, sequencing of mutations associated with reduced partner drug susceptibility will be essential, particularly in those children who had residual day 14 SBP1-parasitemia despite day 7 lumefantrine concentrations exceeding 500 ng/mL (Fig. 4B).”

2) *I would also have liked the authors to explore more the Amplicon sequencing data. is there a change in relative density over time for the persisting or recurrent clones? Do you see any relationship between amplicon sequencing data and 18s or sbp1 data?*

For individual clone density over time, we reported that “the relative abundance of clones could remain stable for several weeks or shift dramatically between clones. In some cases, a clone may remain dominant for a week or two before being replaced by another clone—either a new clone or a previously detected, persistent, low-density clone (Fig. 6C). In other cases, clones would stratify into a dominant clone with several minority clones with relative densities that would remain stable through the end of follow-up (Fig. 6D).”

This pattern was observed in another study over similar time frames.¹ In that study, investigators attempted to construct multi-locus haplotypes. However, using currently available methods, it would be nearly impossible to do this with complex infections ($MOI \geq 3$). In our study, the median MOI was 4.7. As such, we felt that it was appropriate to limit our observations to understanding the MOI over time after treatment and throughout follow-up, as well as assessing broad patterns of clones over time, with a more detailed look into persistent clones within the first 14 days after treatment. As reviewer 2 mentions, it would be ideal to apply mathematical models to the clonal data. This is clearly beyond the scope of this manuscript, but will be conducted once drug resistance and early time points are completed (see Reviewer #1, Response #1 above).

3) *It would be good for the authors to briefly discuss about the limitations of qPCR parasites quantification. Indeed, 18s is a multicopy gene with 5 to 8 copies. On the other sides schizonts can contain 16 to 24 genomes, so the accuracy of the quantification may depend on the relative proportions of the different parasite stages in the sample. Using sbp1 transcripts has also limitations, that may improve specificity by detecting "viable" parasites, but the high number of transcripts may overestimate the real number of rings. It may be the case in this study as the median pre-treatment parasitemia by 18s (2500 parasite/ul) is lower compared to the parasitemia by sbp1 (5000 parasites/ul). This should be the other way around as 18s is supposed to detect the total parasite biomass, and sp1, only ring stages. Using only synchronized ring stages for the standard curve may improve the accuracy of the quantification (similar 18s copies and approximately same number of sp1 transcripts)*

The reviewer is correct that theoretically, 18S quantification should be higher than SBP1 quantification as it is estimating the total parasite biomass. To clarify, our standard curves for quantification used twice synchronized ring-stages for both 18S and SBP1 PCR and we have added this to the methods (lines 438-440):

“This standard curve was created using 10-fold serial dilutions of synchronized ring-stage 3D7 parasites (range 10-10⁻⁸% parasitemia; 5x10⁸-0.5 parasites/mL).”

As for the actual quantification, given that these are estimates of parasite densities in field strains based on a single laboratory strain, we would argue that there is no biologically or statistically significant difference between the parasite densities estimated between 18S and SBP1; as an example, for a given standard curve, the difference between 2500 and 5000 parasites/ μ L is less than 1 PCR cycle, and such a difference also does not have a clinical relevance.

4) *Based on this limitations and figure 2, could the authors speculate on which diagnostic tool would be more accurate in predicting recurrence between 18s and sbp1 assays? Indeed, the probability is between 60 and 80% for 18s, and between 5 and 15% for sbp1 at day 14.*

The answer to this question would largely depend on what “type” of recurrence outcome we want to predict (ie. clinical recurrence of malaria, microscopic recurrence, or recurrence defined some other way (i.e., ring-stages as detected by SBP1)). Additionally, would we want to limit this prediction to new infections or recrudescence? Once the objective is defined, it would again require building a prediction model, which would be very useful, though beyond the scope of this manuscript.

5) *The authors explain the lack of potential protection by TS in the HIV cohort by the small sample size, what about high level of mutations in Pfdhfr and Pfdhps in Uganda potentially associated with resistance to TS?*

This is an important consideration, as antifolate resistance could negate the protective effects of TS. Previous studies in Uganda have demonstrated that the protective effect of TS appears to be maintained even in areas with widespread antifolate resistance.^{2,3} In response to the overall critiques, we have focused this manuscript less on the potential differences between the HIV-infected and HIV-uninfected cohorts, but once resistance genotyping is done (see response #1 above), we will note this in a follow-up paper. It could be an important observation, and we appreciate the reviewer’s comment.

6) *Overall, I find that there are too many references in the results section, it is difficult sometimes to know if the authors are referring to their results or to other publications. I would rather keep the references for the discussion.*

We apologize for this difficulty and have removed most of the references from the results section, except when specifically referring to other studies.

7) *The authors use interchangeably, the terms "persistent infection" or "recurrences". In my understanding, "persistent infections are the ones that do not disappear over time, but recurrences may be "recrudescence infections" or "new infections". The authors should clarify this in the manuscript.*

We believe the reviewer is correct in their interpretation of “persistent infection” and “recurrences.” In accordance with the reviewer’s suggestion, we’ve clarified that recurrences may be either “recrudescence infections” or “new infections” in the intro/background of the manuscript (lines 76-79):

“However, in standard therapeutic efficacy studies, treatment efficacy has been based on monitoring recurrence using microscopy, which has a limit of detection around 50-200 parasites/μL. Detection of microscopic recurrence is typically followed by PCR using size polymorphism-based genotyping to discriminate between recrudescence and new infections.”

8) *Line 59: the reference is for South East Asia, not for sub-Saharan Africa*

The references include both Southeast Asia and sub-Saharan Africa as they are both mentioned in the same sentence.

9) *Line 75: even though NGS techniques are increasingly used in malaria research, old techniques such as agarose gels and capillary electrophoresis are still used, especially in malaria endemic settings*

This is correct, and we’ve revised the sentence to more accurately describe the increasing use of these new techniques (lines 73-76):

“Amplicon deep sequencing, molecular inversion probes, and molecular barcoding are increasingly being used to discriminate and quantify individual clones in multiclonal infections, identify drug resistant variants, and understand the genetic background and evolution of parasite populations.”

10) *Line 79: I would replace "old and new clones" by "recrudescence and new infections"*

We have made the suggested change (line 78-79):

“Detection of microscopic recurrence is followed by PCR using size polymorphism-based genotyping to discriminate between recrudescence and new infections.”

11) *Line 247: What does "sequence" mean? Did you use single SNP or microhaplotypes?*

We sequenced three different markers previously identified to be highly polymorphic for SNPs. Variants, which take into account the entire sequence haplotype (which usually contained many SNPs), were resolved using DADA2. Each “sequence variant” is considered a distinct clone.

12) *Line 415: i would replace "exemplify" by "demonstrate"*

We initially made the change as suggested by the reviewer but ended up removing this statement in the process of shortening the discussion.

Reviewer #2 (Remarks to the Author): Justin Goodwin and colleagues have followed HIV-positive and negative individuals after treatment for *P. falciparum* malaria (with a 3 or 5 day treatment regimen). They screened samples by PCR for parasites during follow-up, used a stage-specific marker to detect ring-stage parasites, measure drug levels, and sequenced parasites during follow-up. They produced a rich dataset that can help to understand treatment failure.

1) *Given the different methods used, and several subgroups analyzed, in parts the text is difficult to follow. Sample sizes are almost never mentioned, not even in the abstract. In the results section, often only percentages are given.*

It would be helpful to add n/N in brackets. Given the many different groups and comparisons, sample sizes end up often relatively small, which makes interpretation of the strength of the results challenging. This is a major weakness of the writing. Following this point, a main question that arose while reading was what this study aims to show in the first place. Is it the comparison of 3d vs 5d treatment? The comparison of HIV+ vs HIV- individuals? A comparison of microscopy, 18S, and SBP1 to determine treatment failure? Clone dynamics? The manuscript would benefit greatly from a clearer focus.

We appreciate the reviewer’s comments, and note that our changes in response have strengthened the manuscript. To address the first point, we have revised the text in many places to add the sample sizes wherever percentages are mentioned. Otherwise, we’ve ensured the sample sizes are included in any relevant tables.

For the second point, we agree with the author that the focus of the study, i.e. whether it was more on the impact of the extended regimen or HIV status, was not always clear throughout the manuscript. We’ve revised the introduction and discussion to emphasize that this manuscript focuses on comparing molecular markers of parasitemia and clonal dynamics between a standard 3-day versus extended 5-day regimen of AL treatment, with less emphasis on differences between HIV infection.

Some of the truly interesting data are deeply hidden in the results and hard to extract. E.g, how many patients with parasites detected by highly sensitive PCR at days 7 and 14 progress to treatment failure, and this should future studies apply these methods to day 7 and 14 samples? And how many samples were available to address these questions?

Of the 303 children in the study, n=211 and n=184 had 18S detected on day 7 and 14, respectively. Of those, n=52 and n=54 went on to fail treatment (defined as recurrent clinical malaria). For SBP1, n=22 and n=43 children had SBP1 mRNA detected on day 7 and 14, respectively. Of those, n=9 and n=13 went on to have recurrent clinical malaria. We also performed logistic regression, adjusting for age, sex, weight, HIV status, treatment regimen, and baseline parasite density. In every model, baseline parasite density was significantly associated with recurrent clinical malaria. Regarding early RT-PCR parasite detection, only 18S detection on day 14 was significantly associated with treatment failure (P = 0.00578) with an odds ratio of 2.77.

2) 18S RT-PCR is well known to be very prone to contamination because of aerosol formation. A large number of negative controls and possibly a cutoff based on CT value is required. Could it be that some of the samples positive by 18S only were false-positive. Please provide details of controls used.

We are happy to provide details of contamination precautions and controls used for RT-PCR experiments. All dried blood spots were cut in a tissue culture hood that was decontaminated with UV light, bleach, then 70% ethanol. All biopsy punches and forceps were decontaminated using three bleach washings followed by three 70% ethanol washings, followed by a blank filter paper punch. Each plate contained a blank control filter paper in a different position on every row (see figure). All steps performed on experimental samples were performed on blank filter paper controls, including extraction, aliquoting, freeze-thawing, plating, and PCR. RT-PCR was performed in a dedicated room in a laminar flow PCR hood. The hood and equipment was cleaned with UV light, bleach, and 70% ethanol before and after every use. Heat labile uracil-DNA glycosylase was included in the

master mix with every PCR run. Each plate included multiple blank filter paper controls and no template controls. Appropriate PPE was worn during each step. Human actin was multiplexed with 18S as another internal extraction control. Standard curves were first validated in triplicate then run in duplicate with every RT-PCR plate and served as positive controls. A conservative CT cutoff threshold of 35 was used based on standard curves.

3) *There is a number of claims that do not seem fully supported by the data. The interpretation of the data is further complicated by the above-mentioned lack of detail, often no P-values or other supporting test results are provided. E.g. lines 190-192: “SBP1-determined recurrence rates were significantly lower during the 14-21, 21-28, and 28-35 day intervals in the extended 5-day regimen as compared to the 3-day regimen.” Please add numbers and a test result. Given the figure the difference seems minimal.*

We would point the reviewer to the referenced table in the text (**Table 1**), which includes all of the relevant information that the reviewer is asking for including point estimations, 95% confidence intervals, P values, and comparison test. The table is referenced in the same sentence that the reviewer points out, however, for clarity we added another reference to the table later on in the same sentence (lines 173-177):

“We found that while 18S-determined recurrence rates were not significantly different between 3-day and 5-day AL regimens (**Table 1** and **Fig. 2B**), SBP1-determined recurrence rates were significantly lower during the 14-21, 21-28, and 28-35 day intervals in the extended 5-day regimen as compared to the 3-day regimen (**Table 1** and **Fig. 2C**).”

Lines 121-122: “Our high-resolution parasitological, clinical, and pharmacological data enabled us to characterize multiclonal infection dynamics at a depth and duration not previously reported’. There is virtually no detail on pharmacological apart from a few mean values in different groups. There are certainly studies with longer follow up and sample size. This statement does not hold.

We disagree with the reviewer regarding the content of pharmacological data, given that we have provided evidence that both the extended drug regimen AND lumefantrine exposure are associated with decreased ring-stage parasitemia after treatment, and demonstrated this in two statistical approaches that use lumefantrine data from nearly 300 children to estimate the predicted probability of ring-stage parasitemia after treatment. However, we have removed this statement so as to not overstate our findings.

Lines 199-201: “parasite densities consistently appeared to be lower in the 5-day regimen compared to the 3-day regimen within the first 28 days after treatment.” In Table 2, hardly any P-values are significant. Given how stochastic density estimates are, I am not convinced to see a clear pattern.

We note that the referenced sentence goes on to state: “...but only parasite densities on day 21 were significantly lower (**Table 2** and **Fig. 3**).” Our data demonstrate that there was a lower density over time in the 5-day treatment group using multiple different methods of detection. This is an interesting and clear pattern that at the very least deserves mention. However, we have softened the language (lines 184-185):

“parasite densities appeared to be lower in the 5-day regimen compared to the 3-day regimen within the first 28 days after treatment...”

Figure 4B: Would it be possible to overlay the raw data (concentrations in each patient)? The curve alone provides little information.

We have adjusted the figure to show the raw day 7 lumefantrine data for all participants with lumefantrine ranges between 0-1000 ng/mL (see the revised figure below). The red points along the top row indicate lumefantrine concentrations for children who did not have day 14 ring-stages detected (n = 179), and the green points show the lumefantrine concentrations of children who had detectable day 14 ring-stages (n = 43). 79 children were not shown on the plot (60 with lumefantrine above 1000 ng/mL and 19 with missing data)—none had ring-stages detected on day 14.

There is a relatively strong focus in title, abstract, and manuscript on clone dynamics. I find this component of the study among the weakest. There are countless mentions of ‘complex patterns’. The main finding of the genotyping is that in some patients, new clones were detected, while in others clones persisted (i.e. true treatment failure), and that many individuals carried multi-clone infections. These data reflects results of many other studies that applied genotyping. I didn’t see a framework to put the genotyping results into a broader picture, despite multiple published mathematical models that do so. Lines 247-262 describe some interesting patterns, but little detail is provided. At the least, please add some numbers on the frequency of each pattern observed. More in-depth analysis would be welcome.

There is a paucity of studies that look at clonal dynamics at multiple time points immediately after treatment, particularly using NGS. Our data allowed us to explore the complexity of infection upon molecular recurrence, the rate of accumulation of clones over time, and see how these differed between a standard and extended ACT regimen, as well as in HIV-infected children taking TS prophylaxis versus HIV-uninfected children living in the same area. We agree that applying a mathematical model to these data would be ideal, but is beyond the scope of this manuscript. Indeed, such a model could benefit from AI prediction, once such methods are available and additional similar datasets in different settings are available. However, per the reviewer’s request, we have added some numbers on the frequency of some of the patterns observed (lines 237-243):

“First, 55% of children with recurrent parasitemia presented with new clones repeatedly throughout follow-up (n = 194 children with ≥ 2 sequenced time points), suggesting a high frequency of new infective mosquito bites and/or the emergence of liver stage parasites between sampling intervals (Fig. 6A). Second, 52% of children with asymptomatic recurrent parasitemia acquired new clones superimposed on previously present clones that had not yet been cleared, such that most recurrent infections consisted of a polyclonal mixture of new and older clones (Fig. 6B).”

The discussion states “We conducted the largest study to date of longitudinal amplicon deep sequencing”. This is another bold claim. I don’t have time to look up every single study and compare sample sizes, but the authors might want to look into <https://pubmed.ncbi.nlm.nih.gov/34146476/>, <https://pubmed.ncbi.nlm.nih.gov/33904907/>, <https://pubmed.ncbi.nlm.nih.gov/31819062/> and others. I don’t think the statement holds.

We agree with the reviewer that this was an overzealous claim. Although we meant specifically in the context of a therapeutic efficacy study, we have removed this statement from the text and make no further claims.

While the authors discuss the limit of detection that allowed them to type a sample (e.g. line 287), the LOD of minority clones in multiple clone infections seems not addressed – potentially more relevant here. The possibility that clones might have been missed at baseline is not mentioned at all, despite multiple modeling approaches to address this problem. These studies should be cited and discussed (and possibly modeling approaches applied to the current data).

We agree with the reviewer that it is important to note the LOD of minority clones. Most studies with high resolution sequencing data set the threshold for calling variants between 1-0.1% of the frequency of total reads per sample, and our study is within this norm (our threshold is a within-host hapotype frequency of 0.1%). It is reasonable to mention this as a limitation although we don't think this warrants a lengthy discussion as this is a limitation to every study that estimates MOI without using those modeling approaches. We have included this in the discussion in lines 347-353:

“A notable limitation to amplicon sequencing is the limit of detection, which is comparable to other nested PCR methods (~ 1,000 parasites/mL), and the practical problem of distinguishing extremely low density clones from sequencing artifacts.^{14,19} As a consequence, sequencing is unable to reliably describe the clonal dynamics of very low density infections, such as may be present in the early post-treatment period, or immediately following parasite emergence from the liver. Furthermore, our data likely underestimates the true MOI, which may be biased towards higher density infections.”

Line 436: “Our molecular stage-specific and clonal parasite data, the largest such dataset published to date”. Of course, this is the largest study using exactly the methods used here, because every study is different. But rather than putting out such statements again and again use the space to cite and discuss previous relevant research.

We agree with the reviewer and have revised the statement to better place our study in the context of other relevant studies (lines 392-393):

“New studies using molecular stage-specific and clonal parasite data, including ours, challenge previously held notions of within-host post-treatment and reinfection dynamics.”

4) At nearly 2500 words, the discussion is extremely long and could easily be shortened. Nearly the full introduction is repeated in the discussion.

We agree with the reviewer and have extensively shortened the discussion by nearly 1000 words. We feel that the discussion is much more focused and readable following these changes.

5) The abstract states that “findings will inform strategies to optimize regimens in the face of emerging artemisinin resistance in Africa”. It would be nice to discuss some of these new strategies in the discussion.

We briefly mention these different strategies in the discussion in the context of some of the limitations of our study. However, we do not expand on them in order to keep the more concise per the reviewer's (and reviewer 3's) request (lines 383-387):

“However, it will be critical to drug resistance genotype our early post-treatment samples (days 1-3) and to compare parasite clearance dynamics using different molecular markers—especially given that extending ACT regimens is among the strategies, along with triple ACT and multiple first-line regimens, being investigated to combat the emergence and spread of antimalarial drug resistance.”

Minor comments:

Line 73-74: In fact (and as mentioned on line 78), size-polymorphic markers have been the primary method to determine MOI in the past

This is correct so we've revised the text to clarify the statement (lines 73-76):

“Amplicon deep sequencing, molecular inversion probes, and molecular barcoding are increasingly being used to discriminate and quantify individual clones in multiclonal infections, identify drug resistant variants, and understand the genetic background and evolution of parasite populations.”

Lines 76-84 are not correct. Many TES have used PCR for well over a decade now.

We respectfully disagree with the reviewer. PCR is typically used for genotype correction after recurrence is identified with microscopy. PCR is also used in human challenge studies. However, PCR has not been used routinely to identify recurrence in TES.

Line 94-97: Please clarify to what studies this statement refers. If the statement refers to treatment efficacy studies, many had large sample sizes and appropriate follow up periods. I do not understand the statement on treatment of asymptomatic individuals.

This statement refers to studies using SBP1 or other ring-stage specific markers. We have clarified this in the text (lines 96-98):

“Molecular studies of ring-stage markers have thus far been limited by small sample sizes, short follow-up duration, or treatment of asymptomatic infections...”

Line 185: Please add a reference.

Reference added.

Lines 192-195 seem to directly contradict lines 185-187. Or do lines 185-187 refer to a different study?

We apologize for the confusion in our writing. Lines 192-195 refer to our molecular analysis using ring-stage parasites as determined by SBP1 RT-PCR. Lines 185-187 refer to the original study, which used microscopy-based outcomes. We've revised the text to make this distinction more clear (lines 171-173):

“We hypothesized that a 5-day AL regimen would significantly reduce molecularly-determined recurrent parasitemia (as defined by 18S or SBP1) compared to a 3-day AL regimen.”

Line 366-368: I would argue follow-up is too short to make this claim.

We agree that the follow-up duration is relatively short; however, we don't believe it's unreasonable to speculate about this in the discussion. But in an effort to reduce the discussion length we have deleted this statement from the manuscript.

Lines 378-379: “In high transmission settings such as ours, differentiating between persistent and recurrent parasitemia in the first few weeks following treatment is challenging.” Why is this, and why does it differ by transmission intensity?

Both Tun et al. and we found a high prevalence of residual 18S rDNA or rRNA within the first 2-3 weeks after treatment in our studies. In our high transmission setting, over 40% of children had microscopically recurrent parasitemia within 28 days. Most of these recurrent parasites were detected in the preceding weeks using molecular methods—the issue arises when trying to genotype such samples using either amplicon deep sequencing or even more traditional size polymorphism-based markers. There is an overlap of residual parasitemia after treatment with recurrent parasitemia that cannot be genotyped due to the low density of these infections. In lower transmission

settings, as in Tun et al., there is a much greater lag time between clearance of residual parasitemia (or molecular markers of parasitemia) and recurrent infection.

Lines 381-385: I believe the wording is wrong. 'Recurrence' can be treatment failure (=same clone) or new infection (=new clone). In line 382, did the authors mean 'early recrudescence'?

In line 382 we specifically mean “recurrence” because we are referring to clones not detected at baseline. This suggests that these are not recrudescing infections, but rather new infections or from parasites present as liver stage merozoites during treatment. As reviewer 3 points out, this does not indicate treatment failure so we avoid using that terminology. But to reviewer 2’s point, we have changed “recurrences” to “new blood-stage infections” to be more explicit (lines 341-342):

“We found that approximately half of these children had new clones not detected at presentation—indicative of potential early new blood-stage infections.”

Are the references in Table S1 supposed to refer to the references in the main manuscript? They don't match.

We apologize for this error and have fixed the references in Table S1 to match the references in the Supplementary text.

Figure 6: Why are no clones shown for Child A at baseline? Given the high parasite density, sequencing seems feasible. Or should line 272 refer to figure 6, not 7?

Line 272 was discussing new and persistent clones in the early post-treatment period, which correctly refers to Figure 7. Figure 6 is showing patterns of clonal dynamics throughout follow-up. Unfortunately, sequencing failed for the day 0 samples for the child in Fig 6A. However, the figure was meant to illustrate how the clones that appeared on the day of failure (day 21) were new and distinct from the clones present on day 14. We’ve revised the figure legend for Fig 6A to include this note.

“**Fig. 6: (A)** A child that progressed to clinical failure on day 21 presented with new clones compared to day 14. The two persistent variants identified with *csp* are discordant with *cpmp* and *cpp*, and thus likely represent different clones that share the same *csp* haplotype. Note that baseline variants are not available for this child due to sequencing failure of pretreatment samples.”

Reviewer #3 (Remarks to the Author): This is a very detailed clinical and parasitological comparison of therapeutic responses to the standard 3 day artemether-lumefantrine antimalarial drug treatment regimen and an extended 5 day course conducted in Ugandan children. The main study was published earlier this year. This is a substantial body of work. These comments (mixing trivial with potentially important) are in manuscript order.

The first sentence of the abstract is rather unwieldy

We’ve simplified the first sentence of the abstract to: “Standard diagnostics used in longitudinal antimalarial studies are unable to characterize the complexity of submicroscopic parasite dynamics, particularly in high transmission settings.”

Line 51: Malaria is a leading cause of morbidity and mortality with over 247 million cases and 619,000 deaths in 2021 – could you add the word “estimated” as we know these figures are inaccurate.

We agree and have made the following change (lines 51-53):

“Malaria is a leading cause of morbidity and mortality with an estimated 249 million cases and 608,000 deaths in 2022, overwhelmingly in sub-Saharan Africa, which contains some of the highest transmission sites in the world.”

Line 58: partial? Odd WHO favoured terminology, but isn't the threat just resistance?

While we agree in principle, we believe the term “partial” artemisinin resistance more accurately describes the phenotype conferred by kelch13 mutations and is in-line with current terminology.

Line 72: with MOI typically measured as 3-8 clones per person (as there are likely other lower density clones).

We agree with this comment and have made incorporated this change (lines 71-73):

“MOI is further thought to scale with transmission intensity, with MOI typically measured as 3-8 clones per person in high transmission settings.”

Line 82: “The use of microscopy is standard in such studies; however, its low sensitivity misses the large submicroscopic burden of infection, which is known to contribute to transmission, drug resistance selection, and recurrent episodes of clinical malaria”. Really? What is the evidence for drug resistance selection? Clinical malaria is associated with microscopy detectable parasitaemias which are preceded by lower densities.

Excellent point by the reviewer. We agree that even though artemisinin and partner drug resistance-associated mutations have been detected in submicroscopic infections, concrete evidence of drug resistance selection in such infections has not been sufficiently demonstrated. Thus, we have revised the statement to indicate that this area needs further study (lines 82-85):

“The use of microscopy is standard in such studies; however, its low sensitivity misses the large submicroscopic burden of infection, which is known to contribute to transmission, asymptomatic and recurrent malaria, and potentially to drug resistance selection and/or spread.”

Line 92: “persistent detection of ring-stage markers that target parasite RNA may be more likely to represent viable circulating asexual parasites”. But this could also represent the well described persistence of development arrested ring stages – which do not necessarily develop further (see Tun et al 2018: 3 days versus 5 days AL -42 days follow up-referenced below), although the measurement of skeleton-binding protein 1 (SBP1) mRNA does suggest potential viability.

We agree with the reviewer, and had inadvertently trimmed this from our submission to meet word limits. Persistent DNA signals could result from “dormant” parasites that do not progress further in the life cycle, and we have revised the intro/background to highlight this phenomenon in the context of persistent parasitemia (lines 91-94) and in the discussion (lines 299-304):

“After treatment, the detection of parasite DNA can be due to circulating gametocytes, residual DNA in the absence of viable asexual parasites, or from artemisinin-induced “dormancy”, metabolically inactive parasites that have undergone developmental arrest.”

“Tun et al. conclude that a sub-population of dormant parasites are the mostly likely explanation for delayed 18S clearance in their study. Artemisinin-induced dormancy is a phenomenon where a subset of asexual parasites temporarily enter a state of arrested growth and metabolic inactivity, which allows them to survive artemisinin treatment. Such dormant parasites may be slowly killed off by partner drugs while contributing to the persistent detection of parasites after treatment.”

Line 150: The limit of parasite detection in this study is not 20 parasites/mL. The actual limit of detection and accuracy of density estimation in the blood spots needs clarification. The calibration curve shows linearity in

diluted test samples to a density below 10 parasites/mL but the samples are dried blood spots, so the limits of detection are determined by the volume of blood sampled, not the assay methodology per-se. Has there been a quantitation validation based on blood spots? More details on sample volume (and filter paper area and thickness and relationship with blood volumes) need to be provided.

We understand the theoretical limit of detecting 1 parasite in a 50 μ L blood sample equates to a limit of detection of 20 parasite/mL. However, detection of extra-erythrocytic 18S in whole blood or plasma likely confounds this number (i.e., the detection of 18S in a blood volume less than 50 μ L). Since our samples were cut from DBS and not from whole blood, we cannot calculate the exact volume of blood used, but it is likely less than 50 μ L. Standard DBS and field DBS were cut using a 6mm in diameter biopsy punch to standardize blood volumes as much as possible (which was validated using human actin). Standard curves consistently detected 50 parasites/mL with stochastic detection of 5 parasite/mL samples. These curves remained linear across all parasite densities sampled (Fig S1). The filter paper area (6mm diameter) and thickness are provided in the methods section. As this is not a methodology paper, we find the validation of standard curves and strong correlation between 18S, SBP1, and microscopy fit for the purpose for this study's objective.

Line 158: "found that 8% and 15% of children had detectable ring-stage parasites on days 7 and 14" – No: had detectable levels of SBP1 mRNA.

We agree and have revised the statement accordingly (lines 140-142):

"Following treatment, we found that 8% (22/286) and 15% (43/283) of children had detectable SBP1 mRNA on days 7 and 14, respectively."

Line 163: On days 7 and 14, the median SBP-1 parasite density was approximately 10,000 parasites/mL (10 parasites/ μ L) -this is very similar to Figure 2 in Tun et al (2018).

We thank the reviewer for pointing this out and have referenced this in the discussion section.

Line 180: were the blood slides rechecked? If still negative - what is your conclusion? The better way to express the relationship between the different parasite density estimation methods is by Bland-Altman plots -not by correlation derived from linear regression analysis.

All slides were read independently by two expert microscopists using the highest standards of QA/QC, though they have not rechecked since the original study was published. It is likely that the slides have degraded enough at this point where it would not be useful to revisit them. There are number of reasons these slides might have been read as negative including variations in staining quality, random variation in the timing of peripheral parasitemia, and human error. We have provided the Bland-Altman plots for the reviewer. The red dashed line is the mean difference line, and the blue dotted lines represent the 95% confidence intervals:

Line 190: *The comparison of the two molecular methods and the difference between the drug regimens is an important observation, suggesting that much of the persistent DNA is not in viable parasites? The exposure response data are also very valuable.*

We agree with the reviewer's conclusion regarding the persistent DNA and have emphasize this more in the revised discussion (lines 315-316):

“Altogether, the discrepancy between 18S and SBP1 suggests that much of the residual 18S rRNA does not represent viable parasites.”

Line 228: *Are the MOI results confounded by parasite density (i.e. the lower the density the lower the characterizable MOI)?*

This is an important consideration, especially when trying to detect clones in low density infections. The following figure shows MOI plotted against parasite density (n = 751). We can see the lower density threshold of 500-1000 parasites/mL, but also a trend towards higher MOI with higher parasite densities. Given the limitations of sequencing low density infections, and the practical limitations of distinguishing very low density sequence variants with sequencing artifacts, we believe it is reasonable that there is likely some bias towards higher MOI in higher density infections. Accordingly, we have included this as a limitation in our study (lines 347-353):

“A notable limitation to amplicon sequencing is the limit of detection, which is comparable to other nested PCR methods (~ 1,000 parasites/mL), and the practical problem of distinguishing extremely low density clones from sequencing artifacts.^{14,19} As a consequence, sequencing is unable to reliably describe the clonal dynamics of very low density infections, such as may be present in the early post-treatment period, or immediately following parasite emergence from the liver. Furthermore, our data likely underestimates the true MOI, which may be biased towards higher density infections.”

Line 285: *“Our findings reveal a striking underestimation of the longitudinal prevalence and density of parasitemia following treatment with the most widely prescribed ACT globally.”-Might be nice to acknowledge Tun et al (2018) who made the very same observation with the same randomized comparison, albeit without detection of persistent parasite mRNA. Importantly Tun et al studied patients in a low transmission setting where reinfection was not a major confounder and there were very few clinical failures. The discussion could be condensed as the CHMI data are not useful in comparison to the clinical trial data.*

We agree with the reviewer and apologize for this omission. We now appropriately acknowledge this study and its observations and have removed the CHMI data in our revised discussion (lines 285-300):

“This global approach painted a picture of high parasite prevalence after treatment, with nearly 65-78% of children with detectable 18S rRNA throughout the entire duration of 6 week follow-up. A previous study which compared the 3 versus 5 day AL regimen was conducted in a low transmission setting in Myanmar, and found a similarly high prevalence and density of 18S rDNA shortly after treatment.⁶ Despite the differences in these studies (asymptomatic vs symptomatic infection, low vs high transmission), the results provide complementary insight into the impact of an extended AL regimen on parasite clearance and recurrence. Notably, neither study found a difference in the prevalence or density of 18S rRNA- or rDNA-determined parasitemia between treatment regimens... Tun et al. conclude that a sub-population of dormant parasites are the mostly likely explanation for delayed 18S clearance in their study.”

Line 340: “Given that ACTs have minimal gametocytocidal effects”; I think minimal is unfair – the activity is substantial. However, this leads to an important point-how do we know that the persistent DNA did not come from gametocytes. Why not look for gametocyte mRNA to answer that definitively?

Although we do clarify that this mainly applies mature gametocytes later on in the sentence, this is a fair point regarding the degree of gametocytocidal effects of ACTs which we address in the revision. We did attempt to look at sex-specific gametocyte dynamics, with particular attention to quantification, but had trouble adapting several specific protocols to our RNA samples. We believe some of these issues were related to their extraction from dried blood spots rather than whole blood stored in an RNA-preservative. Ultimately, we did not have enough individual sample aliquots to do this effectively without compromising samples for other downstream applications due to the number of freeze-thaw cycles required. We are planning such additions in current trials by our group. We’ve revised the text to specify mature gametocytes only, and further mention the lack of gametocyte data as a major limitation of our study (lines 371-381):

“Given that ACTs have minimal gametocytocidal effects against mature gametocytes, a limitation of our study is the lack of gametocyte-specific microscopy or molecular data. It is possible that detectable 18S rRNA and amplicon sequencing markers may have arisen from mature circulating gametocytes that survived treatment.⁷ In the previous 3 versus 5 day AL study in Myanmar, Tun et al. found a high prevalence of persistent 18S rDNA despite the administration of low-dose primaquine.⁶ Furthermore, targeting the ring-stage specific marker SBP1 allowed us to more narrowly assess the asexual parasite burden after treatment, and previous SBP1 studies found no difference in the prevalence, density, or duration of ring-stage parasites with and without gametocytocidal drugs (primaquine or methylene blue).⁷⁻⁹ Nevertheless, additional studies assessing the specific impact of an extended AL regimen on residual gametocytemia are warranted.”

Line 382: If 10 hepatic schizonts liberated their progeny within one day, they would release about 350,000 merozoites into a blood volume of, say, one litre in a young child. That is around the limit of detection in your assay. Please consider that as a possible cause of the “early reinfections”. It does NOT mean drug failure.

We agree and have been careful not to call these “early recurrences” drug failures. Further, we have changed “recurrences” to “new blood-stage infections” to be more explicit (lines 341-342):

“We found that approximately half of these children had new clones not detected at presentation—indicative of potential early new blood-stage infections.”

The 7 pages of discussion are a nice read, but could easily be condensed into 3 without losing the main points.

We agree and have substantially revised and shortened the discussion to be more succinct.

Table 2: suggest either omit or explain it better.

We have moved this table to the supplementary data and revised the table legend to better explain the comparison being shown in the table: “Least squares mean estimated parasite density comparisons between 3 versus 5-day AL.”

Could we have some basic clinical details? fever? anaemia etc?

Clinical details including fever, hemoglobin, WBCs, and liver enzymes were published with the original study. Please see: Whalen ME, et al. The Impact of Extended Treatment With Artemether-lumefantrine on Antimalarial Exposure and Reinfection Risks in Ugandan Children With Uncomplicated Malaria: A Randomized Controlled Trial. *Clin Infect Dis.* 2023 Feb 8;76(3):443-452. doi: 10.1093/cid/ciac783. PMID: 36130191; PMCID: PMC9907485.

AL dosing was weight-based (Coartem □ Dispersible 20 mg/120 mg, Novartis, Switzerland) and administered with milk. Does this mean the dosing was weight adjusted in bands or by exact weight?

Dosing was weight-adjusted in bands. Specifically, participants weighing <15 kg, received 1 tablet; ≥15 to <25 kg, 2 tablets; ≥25 to <35 kg, 3 tablets; and ≥35 kg, 4 tablets. We've added this detail in the material and methods section.

Were the artemether and DHA measurements used in this report?

Only lumefantrine measurements were used in this study, however artemether and DHA data are available and were published in the original study. We plan to incorporate these PK data with parasitological data from earlier time points (days 1-5), in addition to drug resistance data (kelch13 and partner drug mutations) in a future study.

References: Probably does not need all 88 references but you may wish to include and discuss the trial of Tun et al conducted between 2013 and 2015 in Myanmar which also compared 3 days versus 5 days AL, and also documented protracted submicroscopic parasitaemia, but without recrudescence. Tun KM, et al. Effectiveness and safety of 3 and 5 day courses of artemether-lumefantrine for the treatment of uncomplicated falciparum malaria in an area of emerging artemisinin resistance in Myanmar. Malar J. 2018 Jul 11;17(1):258. doi: 10.1186/s12936-018-2404-4. PMID: 29996844; PMCID: PMC6042398.

As noted earlier we have revised the discussion to include the Tun et al. study and regret the original omission. We further cut down on the number of references.

REFERENCES

1. Lerch A, Koepfli C, Hofmann NE, et al. Longitudinal tracking and quantification of individual Plasmodium falciparum clones in complex infections. *Sci Rep*. Mar 4 2019;9(1):3333. doi:10.1038/s41598-019-39656-7
2. Gasasira AF, Kamya MR, Ochong EO, et al. Effect of trimethoprim-sulphamethoxazole on the risk of malaria in HIV-infected Ugandan children living in an area of widespread antifolate resistance. *Malar J*. Jun 23 2010;9:177. doi:10.1186/1475-2875-9-177
3. Homsy J, Dorsey G, Arinaitwe E, et al. Protective efficacy of prolonged co-trimoxazole prophylaxis in HIV-exposed children up to age 4 years for the prevention of malaria in Uganda: a randomised controlled open-label trial. *Lancet Glob Health*. Dec 2014;2(12):e727-36. doi:10.1016/S2214-109X(14)70329-8
4. Lerch A, Koepfli C, Hofmann NE, et al. Development of amplicon deep sequencing markers and data analysis pipeline for genotyping multi-clonal malaria infections. *BMC Genomics*. Nov 13 2017;18(1):864. doi:10.1186/s12864-017-4260-y
5. Jones S, Kay K, Hodel EM, et al. Should Deep-Sequenced Amplicons Become the New Gold Standard for Analyzing Malaria Drug Clinical Trials? *Antimicrob Agents Chemother*. Sep 17 2021;65(10):e0043721. doi:10.1128/AAC.00437-21
6. Tun KM, Jeeyapant A, Myint AH, et al. Effectiveness and safety of 3 and 5 day courses of artemether-lumefantrine for the treatment of uncomplicated falciparum malaria in an area of emerging artemisinin resistance in Myanmar. *Malar J*. Jul 11 2018;17(1):258. doi:10.1186/s12936-018-2404-4
7. Tadesse FG, Lanke K, Nebie I, et al. Molecular Markers for Sensitive Detection of Plasmodium falciparum Asexual Stage Parasites and their Application in a Malaria Clinical Trial. *Am J Trop Med Hyg*. Jul 2017;97(1):188-198. doi:10.4269/ajtmh.16-0893
8. Chang HH, Meibalan E, Zelin J, et al. Persistence of Plasmodium falciparum parasitemia after artemisinin combination therapy: evidence from a randomized trial in Uganda. *Sci Rep*. May 20 2016;6:26330. doi:10.1038/srep26330
9. Mahamar A, Lanke K, Graumans W, et al. Persistence of mRNA indicative of Plasmodium falciparum ring-stage parasites 42 days after artemisinin and non-artemisinin combination therapy in naturally infected Malians. *Malar J*. Jan 9 2021;20(1):34. doi:10.1186/s12936-020-03576-z

Reviewers' Comments:

Reviewer #1:

Remarks to the Author:

I would like to thank the authors for the revised manuscript, most of my comments have been addressed.

However, I still have one comment on the 18s RTqPCR. The limitation of the assay is not fully described in the discussion. It is true that the assay may detect gametocyte and free circulating DNA, but the quantification is not absolute, as it will depend the standard curve used (here with synchronized rings, and the different parasite stages in the patients' samples. As mentioned in my previous comment, the estimated parasite density by 18s RTqPCR should be higher than the one estimated by SBP1, especially as your 18s assay is using total nucleic acid, the density found should be at least 1 log higher compared to microscopy, due to the fact that there are different stages such as schizonts with multiple genomes.

For example, when looking at table S2 for baseline parasite density, the lowest density is with 18s and the highest with microscopy. SBP1 being in the middle. This should be the opposite, as 18s RTqPCR is more sensitive, and detecting gametocyte and circulating free DNA, and the difference between both is quite significant.

Additionally, there is discrepancy between Figure 1b and Figure 3. Figure 1 is concordant with Table S1, but Figure 3 does make more sense, as the parasite density with 18s RTqPCR is higher compared to microscopy, and SBP1 is between both.

Minor comments

- Lines 383-384: Please revise the sentence : "However, it will be critical to drug resistance genotype our early post-treatment samples,..."
- Line 398: I do not understand the sentence: "Although this did not appear to impact clinical outcomes, this time period is marked by declining levels of partner drug monotherapy." Patients were treated with ACTs, not with monotherapies ?

Reviewer #2:

Remarks to the Author:

The authors have submitted a much improved and clarified manuscript. A number of major and minor points remain.

Major points

1) Still referring to what the main message(s) of this study are: Lines 123-136 show that 205/291 children develop parasitemia within 42 days after treatment (by PCR), 50% were also positive by microscopy. This points to either massive treatment failure by day 42, or many new infections. My immediate question seeing these numbers is how many of these children carry the same clone as at baseline.

Yet, the sequencing data focusses on a much smaller subset of 20 children within two weeks after treatment (lines 250-266). I believe comparing clones at baseline to those during follow up would be much more important than what is described in lines 233-248, these lines essentially sum up patterns of clone dynamics observed in other asymptomatic cohorts before (on a side note, I disagree with the response of the authors that such studies do not exist, cohorts/models including clone detectability, recurrence vs. probability of new infection with the same clone, etc. have been published. But I agree that this might go beyond the scope of this manuscript).

Once these data are added, the authors could be much more explicit in the discussion on whether they believe to see much higher rates of treatment failure than currently being reported. This point remains vague in the discussion.

2) Lines 156/169 and Figure 1: It is impossible that ring stage density (5000/uL) is higher than all parasite density (2500/uL). These conflicting data point to a problem in quantification.

Minor points

Line 167: underestimated is the wrong word here. Maybe 'missed'?

Lines 184-186: These data indicate a problem with the microscopy applied. Are such rates of false-negative high-density infections normal? Maybe discuss in a bit more detail.

Figure 1: Y-axis label is cut off in panels B and D

Figure 2: The selection of color and shading makes it very hard to read this figure

Figure 4A shows two numbers – I don't think these data warrant a figure.

Line 286: I believe the word 'nearly' can be removed

Line 342 and 397: While somewhat covered in the paragraph following line 342, the authors could state more clearly that even in pre-treatment samples it is possible that some clones were not detected. See e.g. <https://pubmed.ncbi.nlm.nih.gov/30833657/> for some data on detectability of clones in ampseq data (though that study sequenced asymptomatic infections).

Line 383: I am not sure the wording "to drug resistance genotype" exists. Maybe write "sequence markers of drug resistance"

Line 790: I believe you mean 'parasitemia by microscopy'. Obviously all the children in the plots carry parasites by PCR, else they wouldn't appear in the plot.

Reviewer #3:

Remarks to the Author:

The authors have addressed the main points raised by the reviewers and revised their paper accordingly. I have no further comments or concerns.

Reviewer #1:

I would like to thank the authors for the revised manuscript, most of my comments have been addressed.

1) However, I still have one comment on the 18s RTqPCR. The limitation of the assay is not fully described in the discussion. It is true that the assay may detect gametocyte and free circulating DNA, but the quantification is not absolute, as it will depend the standard curve used (here with synchronized rings, and the different parasite stages in the patients' samples. As mentioned in my previous comment, the estimated parasite density by 18s RTqPCR should be higher than the one estimated by SBP1, especially as your 18s assay is using total nucleic acid, the density found should be at least 1 log higher compared to microscopy, due to the fact that there are different stages such as schizonts with multiple genomes. For example, when looking at table S2 for baseline parasite density, the lowest density is with 18s and the highest with microscopy. SBP1 being in the middle. This should be the opposite, as 18s RTqPCR is more sensitive, and detecting gametocyte and circulating free DNA, and the difference between both is quite significant.

We understand the reviewer's point regarding the quantification of parasites in our study, and agree that, theoretically, 18S-determined parasite density should be greater than SBP1 ring-stage density. Any study that attempts to quantify parasite density depends on the standards used to calculate the final density, whether it be absolute copy number of plasmids/DNA standards, or estimates based on cultured parasite densities. Further, each of these methods are estimates of the true density, and it is not possible (at least with our study design) to say whether 18S underestimates total parasite density or if SBP1 overestimates ring-stage density. What is most critical is that all of our methods all highly correlated and in agreement with one another. Indeed, many methodological studies estimating parasite density have similar results. Nwakanma et al. found that 18S qPCR had a mean \log_{10}/mL difference of 0.41 less than microscopy, while Kamau et al found an 18S qPCR mean \log_{10}/mL difference of 0.87 less than microscopy.^{1,2} Many other well validated studies have 18S parasite densities greater than microscopy,³ while others estimate parasite densities greater than or less than microscopy (in the same study), depending on if they use 18S rRNA or 18S rDNA.⁴ In fact, there is only one other study performing absolute quantification of SBP1 which found a \log_{10}/mL ring-stage density 0.77 greater than microscopy.⁵ To put our study in perspective, the \log_{10}/mL difference between 18S and microscopy was 0.6, and between 18S and SBP1 was 0.4, well within the normal range published in methodological papers. Again, however, it is impossible to discern what the true parasite density is, and whether these studies are either underestimating 18S parasite density, or overestimating microscopy.

What is most important about our results is how well each of these independent methods of estimating parasite density correlate with one another. For example, two well validated markers of estimating parasite density, varATS and TARE-2, were found to have a correlation of 0.68 and 0.66 with microscopy, respectively.³ In that same highly referenced study, 18S was found to have a correlation of 0.74 with microscopy (all Pearson's correlations).³ In our study, the correlation of 18S and SBP1 with microscopy was 0.82 and 0.85, respectively. The correlation of 18S and SBP1 with each other was 0.89. Although we report the Spearman values in Fig. S2, the values reported here are Pearson's correlations for direct comparison. With the above in mind, we definitely agree with the reviewer that the limitation of the assays is not fully described in the discussion and should be spelled out plainly. We have revised the discussion to include this limitation (lines 315-320):

"Notably, SBP1-estimated ring-stage parasite densities were slightly higher than the total 18S-estimated parasite densities, highlighting an inherent limitation of parasite quantification based on standard curves derived from a single laboratory strain—which is unlikely to capture the diversity and variable gene expression of field strains. Reassuringly, the strong correlation between our microscopy, 18S, and SBP1 estimated parasite densities demonstrates the robustness of our comparative analysis (Fig. S2)."

2) Additionally, there is discrepancy between Figure 1b and Figure 3. Figure 1 is concordant with Table S1, but Figure 3 does make more sense, as the parasite density with 18s RTqPCR is higher compared to microscopy, and SBP1 is between both.

We appreciate the comment from the reviewer, and it is due to a slight misinterpretation of what each figure represents, but necessitates improved clarity in our figure legend. In Fig. 1 we are showing raw parasite densities for each method of detection. Thus, the lower mean and median densities of 18S < SBP1 < microscopy are a

consequence of the better sensitivity of the molecular assays, with 18S having many more low-density infections detected than SBP1 and microscopy, which decreases the mean and median density (day 14 is a good example). However, *in Fig. 3*, we provide least squares means estimates of parasite densities, as opposed to the raw values reported in Fig 1. This uses a statistical model (based on the raw data) that adjusts for covariates such as treatment group, HIV status, and an interaction between treatment group and time, to describe the impact of treatment regimen on parasite density throughout follow-up. We have revised the Fig 3. figure legend to make this more clear (lines 21-24):

“Fig. 3 | Estimated parasite densities after treatment based on standard (3-day) or extended (5-day) AL. Least squares means estimated parasite densities determined by (A) microscopy, (B) 18S, or (C) SBP1 parasitemia. Estimates are adjusted for treatment regimen, time as categorical variable, an interaction product of treatment group and time, and log₁₀ transformed baseline parasite density. The p-values for HIV status and the interaction between HIV status and AL group were not significant; therefore, they were not adjusted for in the final model.”

Minor comments

3) Lines 383-384: Please revise the sentence : "However, it will be critical to drug resistance genotype our early post-treatment samples,..."

We have revised the sentence to the following (lines 389-390):

“However, it will be critical to genotype our early post-treatment samples (days 1-3) for drug resistance-associated mutations, and...”

4) Line 398: I do not understand the sentence: "Although this did not appear to impact clinical outcomes, this time period is marked by declining levels of partner drug monotherapy." Patients were treated with ACTs, not with monotherapies?

We mean to state that in this post-treatment period (day 14 onward), patients only have residual partner drug in their plasma, as all the artemisinin and DHA has been eliminated. This essentially makes the treatment a “monotherapy” of only partner drug after the first several days of therapy are completed. We appreciate that this was unclear, and have revised the sentence to be more explicit (lines 404-405):

“Although this did not appear to impact clinical outcomes, parasites are only exposed to declining levels of partner drug throughout this time.”

Reviewer #2 (Remarks to the Author):

The authors have submitted a much improved and clarified manuscript. A number of major and minor points remain.

Major points

1) Still referring to what the main message(s) of this study are: Lines 123-136 show that 205/291 children develop parasitemia within 42 days after treatment (by PCR), 50% were also positive by microscopy. This points to either massive treatment failure by day 42, or many new infections. My immediate question seeing these numbers is how many of these children carry the same clone as at baseline. Yet, the sequencing data focusses on a much smaller subset of 20 children within two weeks after treatment (lines 250-266). I believe comparing clones at baseline to those during follow up would be much more important than what is described in lines 233-248, these lines essentially sum up patterns of clone dynamics observed in other asymptomatic cohorts before (*on a side note, I disagree with the response of the authors that such studies do not exist, cohorts/models including clone detectability, recurrence vs. probability of new infection with the same clone, etc. have been published. But I agree that this might go beyond the scope of this manuscript*). Once these data are added, the authors could be much more explicit in the discussion on whether they believe to see much higher rates of treatment failure than currently being reported. This point remains vague in the discussion.

We do not believe that the results of this study point to massive treatment failure, but rather suggest many new infections occurring over time in this high transmission setting where individuals are continuously exposed to infectious mosquito bites. In the discussion we note: “Notably, most persistent clones were not detected beyond 14 days, potentially reflecting the sustained impact of lumefantrine given its half-life of 3-4 days.” This is supported by PCR correction of treatment failures in the original study (the current WHO standard), which demonstrated that AL remains efficacious and suggests that the overwhelming majority of microscopically recurrent parasitemia represents new infections, not recrudescence.

While it would be interesting to perform amplicon sequencing for genotypic correction of treatment failures (comparing every detectable clone at baseline to those detectable during follow-up), this has not been approved by the WHO as a method of determining treatment efficacy, and we are unaware of any therapeutic efficacy studies (TES) to date that have used amplicon sequencing as a primary endpoint for determining treatment efficacy.^{6,7} Evaluating the use of next generation sequencing (NGS) for genotyping is not the purpose of this manuscript, and would require additional methodological rigor - most notably, assays should be performed in triplicate with stricter criteria for determining sequencing errors, as was performed in Gruenberg et al. in a much smaller set of samples (n=34 paired samples) in order to avoid overestimating treatment failures. The purpose of our study was not to use NGS for genotypic correction, thus sequencing was performed with criteria chosen to maximize the detectability of clones over time, and includes over 2000 total samples. As such, it would not be appropriate to estimate treatment failures using NGS with the methodology utilized in our study. That being said, we agree with the reviewer that more clarity was needed in our discussion as to the goal of the amplicon-based sequencing. We have revised the concluding paragraph to make this much clearer (lines 399-413):

“We demonstrate the surprising prevalence of post-treatment parasitemia and the impact of partner drug exposure on circulating ring-stage parasites. Multilocus amplicon sequencing of persistent ring-stage parasites on day 14 further revealed that nearly half of the clones were present prior to treatment, while the rest appeared to be very early new infections. Although this did not appear to impact clinical outcomes, parasites are only exposed to declining levels of partner drug throughout this time. Previous *m*sp1, *m*sp2, and microsatellite genotyping demonstrated that the overwhelming majority of microscopically recurrent parasitemia consisted of new infections, not recrudescence.⁸ This suggests that the enormous burden of post-treatment parasitemia seen in our study is reflective of a high rate of new infections rather than treatment failures—and of the overall continued efficacy of AL. However, the possibility of residual ring-stage parasites, combined with early new infections, may have ramifications for the emergence and spread of ACT resistance, especially as partial artemisinin resistance currently threatens multiple African countries already facing widespread mutations conferring reduced partner drug susceptibility.^{9-11”}

2) Lines 156/169 and Figure 1: It is impossible that ring stage density (5000/uL) is higher than all parasite density (2500/uL). These conflicting data point to a problem in quantification.

Please see the detailed response given under Reviewer 1, Response 1.

Minor points

3) Line 167: underestimated is the wrong word here. Maybe ‘missed’?

We have revised the wording as follows (line 286):

“Overall, microscopy failed to detect 39% (215/553) of the total prevalence of ring-stage parasites...”

6) Lines 184-186: These data indicate a problem with the microscopy applied. Are such rates of false-negative high-density infections normal? Maybe discuss in a bit more detail.

Rather than indicating a problem with our applied microscopy, this indicates a limitation of the technique in general. It is well known that microscopy underestimates the true prevalence of parasites. In studies that have stratified this by parasite density, there is a strong correlation with decreasing parasite density and the number of false

negatives.^{12,13} In a recent paper by Opoku Afriyie et al., their WHO-certified microscopists failed to detect parasite densities between 100-999 parasites/ μ L over 63% of the time, and failed to detect higher parasite densities between 1000-4999 parasites/ μ L over 34% of the time.¹³ All of our slides were read independently by two Level 3 Integrated Quality Laboratory Services (IQLS) rated microscopists (level 4 is expert). That our microscopists missed ~20% of PCR positive samples is well within the known limitations of microscopy. Furthermore, a limit of detection of 100 parasites/ μ L describes the limits of detection in the most ideal field conditions, which is often not the case.

7) Figure 1: Y-axis label is cut off in panels B and D

We thank the reviewer for pointing this out and have corrected the Y-axis label.

8) Figure 2: The selection of color and shading makes it very hard to read this figure

We apologize for this. We maintained the color selection for consistency across figures; however, we have increased the size of the lines and shading of the confidence intervals so that they stand out more.

9) Figure 4A shows two numbers – I don't think these data warrant a figure.

We believe that this figure emphasizes an important finding in the text and would prefer to keep it.

10) Line 286: I believe the word 'nearly' can be removed

Removed.

11) Line 342 and 397: While somewhat covered in the paragraph following line 342, the authors could state more clearly that even in pre-treatment samples it is possible that some clones were not detected. See e.g. <https://pubmed.ncbi.nlm.nih.gov/30833657/> for some data on detectability of clones in ampseq data (though that study sequenced asymptomatic infections).

Thank you to the reviewer for pointing this out; it is true that not only can low density infections be missed, but low-density clones can also be missed, particularly in pre-treatment samples, resulting in misclassification of clones. We have added a statement noting this limitation in the discussion section pointed out by the reviewer (lines 357-358):

“Low-density clones present prior to treatment may have also been missed, resulting in persistent clones being misclassified as new infections.”

12) Line 383: I am not sure the wording “to drug resistance genotype” exists. Maybe write “sequence markers of drug resistance”

We have revised the sentence to the following (lines 389-390):

“However, it will be critical to genotype our early post-treatment samples (days 1-3) for drug resistance-associated mutations, and...”

13) Line 790: I believe you mean 'parasitemia by microscopy'. Obviously all the children in the plots carry parasites by PCR, else they wouldn't appear in the plot.

This is correct, as the WHO only uses microscopy to define treatment outcomes. We have made the change to the manuscript for clarity (lines 11-12):

“ACPR is defined as an absence of microscopic parasitemia irrespective of fever throughout follow-up.”

Reviewer #3 (Remarks to the Author):

The authors have addressed the main points raised by the reviewers and revised their paper accordingly. I have no further comments or concerns.

References

1. Nwakanma DC, Gomez-Escobar N, Walther M, et al. Quantitative detection of Plasmodium falciparum DNA in saliva, blood, and urine. *J Infect Dis*. Jun 1 2009;199(11):1567-74. doi:10.1086/598856
2. Kamau E, Alemayehu S, Feghali KC, Saunders D, Ockenhouse CF. Multiplex qPCR for detection and absolute quantification of malaria. *PLoS One*. 2013;8(8):e71539. doi:10.1371/journal.pone.0071539
3. Hofmann N, Mwingira F, Shekalaghe S, Robinson LJ, Mueller I, Felger I. Ultra-sensitive detection of Plasmodium falciparum by amplification of multi-copy subtelomeric targets. *PLoS Med*. Mar 2015;12(3):e1001788. doi:10.1371/journal.pmed.1001788
4. Tadesse FG, Lanke K, Nebie I, et al. Molecular Markers for Sensitive Detection of Plasmodium falciparum Asexual Stage Parasites and their Application in a Malaria Clinical Trial. *Am J Trop Med Hyg*. Jul 2017;97(1):188-198. doi:10.4269/ajtmh.16-0893
5. Mahamar A, Lanke K, Graumans W, et al. Persistence of mRNA indicative of Plasmodium falciparum ring-stage parasites 42 days after artemisinin and non-artemisinin combination therapy in naturally infected Malians. *Malar J*. Jan 9 2021;20(1):34. doi:10.1186/s12936-020-03576-z
6. Gruenberg M, Lerch A, Beck HP, Felger I. Amplicon deep sequencing improves Plasmodium falciparum genotyping in clinical trials of antimalarial drugs. *Sci Rep*. Nov 28 2019;9(1):17790. doi:10.1038/s41598-019-54203-0
7. Jones S, Kay K, Hodel EM, et al. Should Deep-Sequenced Amplicons Become the New Gold Standard for Analyzing Malaria Drug Clinical Trials? *Antimicrob Agents Chemother*. Sep 17 2021;65(10):e0043721. doi:10.1128/AAC.00437-21
8. Whalen ME, Kajubi R, Goodwin J, et al. The Impact of Extended Treatment With Artemether-lumefantrine on Antimalarial Exposure and Reinfection Risks in Ugandan Children With Uncomplicated Malaria: A Randomized Controlled Trial. *Clin Infect Dis*. Feb 8 2023;76(3):443-452. doi:10.1093/cid/ciac783
9. Conrad MD, Asua V, Garg S, et al. Evolution of Partial Resistance to Artemisinins in Malaria Parasites in Uganda. *N Engl J Med*. Aug 24 2023;389(8):722-732. doi:10.1056/NEJMoa2211803
10. Uwimana A, Legrand E, Stokes BH, et al. Emergence and clonal expansion of in vitro artemisinin-resistant Plasmodium falciparum kelch13 R561H mutant parasites in Rwanda. *Nat Med*. Oct 2020;26(10):1602-1608. doi:10.1038/s41591-020-1005-2
11. Ehrlich HY, Bei AK, Weinberger DM, Warren JL, Parikh S. Mapping partner drug resistance to guide antimalarial combination therapy policies in sub-Saharan Africa. *Proc Natl Acad Sci U S A*. Jul 20 2021;118(29):doi:10.1073/pnas.2100685118
12. McKenzie FE, Sirichaisinthop J, Miller RS, Gasser RA, Jr., Wongsrichanalai C. Dependence of malaria detection and species diagnosis by microscopy on parasite density. *Am J Trop Med Hyg*. Oct 2003;69(4):372-6.
13. Opoku Afriyie S, Addison TK, Gebre Y, et al. Accuracy of diagnosis among clinical malaria patients: comparing microscopy, RDT and a highly sensitive quantitative PCR looking at the implications for submicroscopic infections. *Malar J*. Mar 4 2023;22(1):76. doi:10.1186/s12936-023-04506-5

Reviewers' Comments:

Reviewer #1:

Remarks to the Author:

I'm happy with the revised version, I have no further comment.

Reviewer #2:

Remarks to the Author:

The authors provided a well thought-through and further clarified manuscript. I have no further comments.

Cristian Koepfli